# CoBERL: Contrastive BERT for Reinforcement Learning

Andrea Banino [1,+]                 Adrià Puidomenech Badia [1,+]                 Jacob Walker [1,+]

Tim Scholtes [1,+]                 Jovana Mitrovic [1]                 Charles Blundell [1]

## Abstract

Many reinforcement learning (RL) agents require a large amount of experience to solve tasks. We propose Contrastive BERT for RL (CoBERL), an agent that combines a new contrastive loss and a hybrid LSTM-transformer architecture to tackle the challenge of improving data efficiency. CoBERL enables efficient and robust learning from pixels across a wide variety of domains. We use bidirectional masked prediction in combination with a generalization of a recent contrastive method to learn better representations for RL, without the need of hand engineered data augmentations. We find that CoBERL consistently improves data efficiency across the full Atari suite, a set of control tasks and a challenging 3D environment, and often it also increases final score performance.

## 1 Introduction

Developing sample efficient reinforcement learning (RL) agents that only rely on raw high dimensional inputs is challenging. Specifically, it is difficult as it often requires us to simultaneously train several neural networks based on sparse environment feedback and strong correlation between consecutive observations. This problem is particularly severe when the networks are large and densely connected, like in the case of the transformer (Vaswani et al., 2017) due to noisy gradients often found in RL problems. Outside of the RL domain, transformers have proven to be very expressive (Brown et al., 2020) and, as such, they are of particular interest for complex domains like RL.

In this paper, we propose to tackle this shortcoming by taking inspiration from Bidirectional Encoder Representations for Transformers (BERT; Devlin et al., 2019), and its successes on difficult sequence prediction and reasoning tasks. Specifically we propose a novel agent, named Contrastive BERT for RL (CoBERL), that combines a new contrastive representation learning objective with architectural improvements that effectively combine LSTMs with transformers.

For representation learning we take inspiration from previous work showing that contrastive objectives improve the performance of agents (Fortunato et al., 2019; Srinivas et al., 2020b; Kostrikov et al., 2020; Mitrovic et al., 2021). Specifically, we combine the paradigm of masked prediction from BERT (Devlin et al., 2019) with the contrastive approach of Representation Learning via Invariant Causal Mechanisms (ReLIC; Mitrovic et al., 2021). Extending the BERT masked prediction to RL is not trivial; unlike in language tasks, there are no discrete targets in RL. To circumvent this issue, we extend ReLIC to the time domain and use it as a proxy supervision signal for the masked prediction. Such a signal aims to learn self-attention-consistent representations that contain the appropriate information for the agent to effectively incorporate previously observed knowledge in the transformer weights. Critically, this objective can be applied to different RL domains as it does not require any data augmentations and thus circumvents the need for much domain knowledge. This is another advantage compared to the original ReLIC and which require to hand-design augmentations.

In terms of architecture, we base CoBERL on Gated Transformer-XL (GTrXL; Parisotto et al., 2020) and Long-Short Term Memories (LSTMs; Hochreiter & Schmidhuber, 1997). GTrXL is an adaptation of a transformer architecture specific for RL domains. We combine GTrXL with LSTMs using a gate trained via an RL loss. This allows the agent to learn to exploit the representations

---

[01] DeepMind London, [+] Equal Contribution. Corresponding author: abanino@deepmind.com

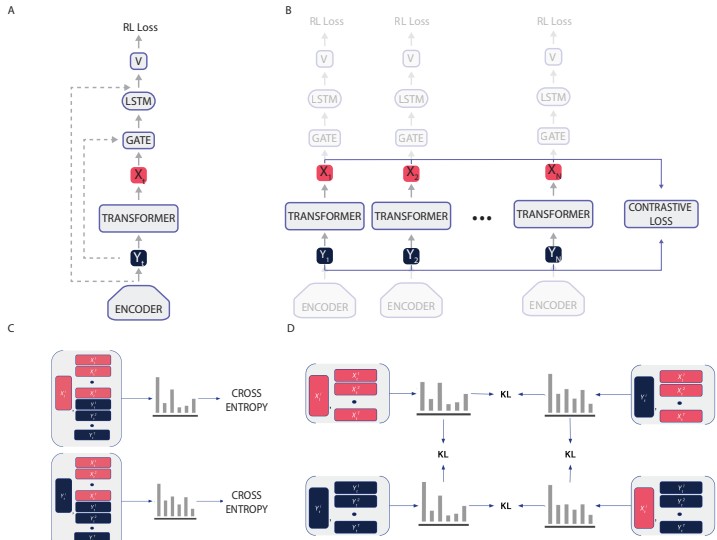

Figure 1: COBERL. A) General Architecture. We use a residual network to encode observations into embeddings $Y_t$. We feed $Y_t$ through a causally masked GTrXL transformer, which computes the predicted masked inputs $X_t$ and passes those together with $Y_t$ to a learnt gate. The output of the gate is passed through a single LSTM layer to produce the values that we use for computing the RL loss. B) Contrastive loss. We also compute a contrastive loss using predicted masked inputs $X_t$ and $Y_t$ as targets. For this, we do not use the causal mask of the Transfomer. For details about the contrastive loss, please see Section 2.1. C) Computation of $q_x^t$ and $q_y^t$ (from Equation 1) with the round brackets denoting the computation of the similarity between the entries D) Regularization terms from Eq. 3 which explicitly enforce self-attention consistency.

offered by the transformer only when an environment requires it, and avoid the extra complexity when not needed.

We extensively test our proposed agent across a widely varied set of environments and tasks ranging from 2D platform games to 3D first-person and third-person view tasks. Specifically, we test it in the control domain using DeepMind Control Suite (Tassa et al., 2018) and probe its memory abilities using DMLab-30 (Beattie et al., 2016). We also test our agent on all 57 Atari games (Bellemare et al., 2013). Our main contributions are:

- A novel contrastive representation learning objective that combines the masked prediction from BERT with a generalization of RELIC to the time domain; with this we learn self-attention consistent representations and extend BERT-like training to RL using contrastive objectives without the need of hand-engineered augmentations.

- An improved architecture that, using a gate, allows us to flexibly integrate knowledge from both the transformer and the LSTM.

- Improved data efficiency (and in some cases also overall performance) on a varied set of environments. Also, we show that individually both our contrastive loss and the architecture improvements play a role in improving performance.

## 2 METHOD

To tackle the problem of data efficiency in deep reinforcement learning, we propose two modifications to the status quo. First, we introduce a novel representation learning objective aimed at learning better representations by enforcing self-attention consistency in the prediction of masked inputs. Second, we propose an architectural improvement to combine the strength of LSTMs and transformers.

### 2.1 REPRESENTATION LEARNING IN REINFORCEMENT LEARNING

It has been empirically observed that performing RL directly from high dimensional observations (e.g. raw pixels) is sample inefficient (Lake et al., 2017); however, there is evidence that learning from low-dimensional state based features is significantly more efficient (e.g. Tassa et al., 2018). Thus,

if state information could be effectively extracted from raw observations it may then be possible to learn from these as fast as from states. However, unlike in the supervised or unsupervised setting, learning representations in RL is complicated by non-stationarity in the training distribution and strong correlation between observations at adjacent timesteps. Furthermore, given the often sparse reward signal coming from the environment, learning representations in RL has to be achieved with little to no supervision. Approaches to address these issues can be broadly classified into two classes. The first class uses auxiliary self-supervised losses to accelerate the learning speed in model-free RL algorithms (Schmidhuber, 1990; Jaderberg et al., 2016; Oord et al., 2018; Srinivas et al., 2020a). The second class learns a world model and uses this to collect imagined rollouts, which then act as extra data to train the RL algorithm reducing the samples required from the environment (Sutton, 1990; Ha & Schmidhuber, 2018; Kaiser et al., 2019; Schrittwieser et al., 2020). CoBERL is part of the first set of methods, as it uses a self-supervised loss to improve data efficiency. In particular, we take inspiration from the striking progresses made in recent years in both masked language modelling (BERT, Devlin et al., 2019) and contrastive learning (Oord et al., 2018; Chen et al., 2020).

From BERT we borrow the combination of bidirectional processing in transformers (rather than left-to-right or right-to-left, as is common with RNN-based models such as LSTMs) with a masked prediction setup. With this combination the model is forced to focus on the context provided by the surrounding timesteps to solve its objective (Voita et al., 2019). Thus, when the model is asked to reconstruct a particular masked frame it does so by attending to all the relevant frames in the trajectory. This is of particular relevance in RL, where state aliasing is a source of uncertainty in value estimation, and so we believe that attending to other frames in the sequence could be a way to provide extra evidence to reduce this uncertainty.

However, unlike in BERT where the input is a discrete vocabulary for language learning and targets are available, in RL inputs consist of images, rewards and actions that do not form a finite or discrete set and targets are not available. Thus, we must construct proxy targets and the corresponding proxy tasks to solve. For this we use contrastive learning, and we derive our contrastive loss from RELIC (Mitrovic et al., 2021). RELIC creates a series of augmentations from the original data and then enforces invariant prediction of proxy targets across augmentations through an invariance regularizer, yielding improved generalization guarantees. Compared to RELIC, CoBERL does not use data augmentations. Instead we rely on the sequential nature of our input data to create the necessary groupings of similar and dissimilar points needed for contrastive learning. Not having a need for augmentations is critical for RL as each domain would require handcrafting different augmentations. Augmentations also make other methods less data efficient. Finally, we do not use an additional encoder network as in RELIC, thus making CoBERL fully end-to-end.

We now set out to explain the details of the auxiliary loss. In a batch of sampled sequences, before feeding embeddings into the transformer stack, $15\%$ of the embeddings are replaced with a fixed token denoting masking. Then, let the set $\mathcal{T}$ represent indices in the sequence that have been randomly masked and let $t \in \mathcal{T}$. For the $i$-th training sequence in the batch, for each index $t \in \mathcal{T}$, let $x_t^i$ be the output of the GTrXL and $y_t^i$ the corresponding input to the GTrXL from the encoder (see Fig 1B). Let $\phi(\cdot, \cdot)$ be the inner product defined on the space of critic embeddings, i.e. $\phi(x, y) = g(x)^T g(y)$, where $g$ is a critic function. The critic separates the embeddings used for the contrastive proxy task and the downstream RL task. Details of the critic function are in App. B. This separation is needed since the proxy and downstream tasks are related but not identical, and as such the appropriate representations will likely not be the same. As a side benefit, the critic can be used to reduce the dimensionality of the dot-product. To learn the embedding $x_t^i$ at mask locations $t$, we use $y_t^i$ as a positive example and the sets $\{y_t^b\}_{b=0, b \neq i}^B$ and $\{x_t^b\}_{b=0, b \neq i}^B$ as the negative examples with $B$ the number of sequences in the minibatch. We model $q_x^t$ as

$$q_x^t = \frac{\exp(\phi(x_t^i, y_t^i))}{\sum_{b=0}^B \exp(\phi(x_t^i, y_t^b)) + \exp(\phi(x_t^i, x_t^b))} \tag{1}$$

with $q_x^t$ denoting $q_x^t \left( x_t^i | \{y_t^b\}_{b=0}^B, \{x_t^b\}_{b=0, b \neq i}^B \right)$; $q_y^t$ is computed analogously (see Fig 1C). In order to enforce self-attention consistency in the learned representations, we explicitly regularize the similarity between the pairs of transformer embeddings and inputs through Kullback-Leibler regularization from RELIC. Specifically, we look at the similarity between appropriate embeddings and inputs, and within the sets of embeddings and inputs separately. To this end, we define:

$$p_x^t = \frac{\exp(\phi(x_t^i, y_t^i))}{\sum_{b=0}^{B} \exp(\phi(x_t^i, y_t^b))}; \quad s_x^t = \frac{\exp(\phi(x_t^i, x_t^i))}{\sum_{b=0}^{B} \exp(\phi(x_t^i, x_t^b))} \tag{2}$$

with $p_x^t$ and $s_x^t$ shorthand for $p_x^t(x_t^i | \{y_t^b\}_{b=0}^B)$ and $s_x^t(x_t^i | \{x_t^b\}_{b=0}^B)$, respectively; $p_y^t(y_t^i | \{x_t^b\}_{b=0}^B)$ and $s_y^t(y_t^i | \{y_t^b\}_{b=0}^B)$ defined analogously (see Fig 1D). All together, the final objective takes the form:

$$\mathcal{L}(X, Y) = -\sum_{t \in \mathcal{T}} (\log q_x^t + \log q_y^t) + \alpha \sum_{t \in \mathcal{T}} \Big[ KL(s_x^t, \text{sg}(s_y^t)) + KL(p_x^t, \text{sg}(p_y^t)) +$$
$$KL(p_x^t, \text{sg}(s_y^t)) + KL(p_y^t, \text{sg}(s_x^t)) \Big] \tag{3}$$

with $\text{sg}(\cdot)$ indicating a stop-gradient. As in our RL objective, we use the full batch of sequences that are sampled from the replay buffer to optimize this contrastive objective. Finally, we optimize a weighted sum of the RL objective and $\mathcal{L}(X, Y)$ (see App. C for the details on the weighting).

## 2.2 ARCHITECTURE OF COBERL.

While transformers have proven very effective at connecting long-range data dependencies in natural language processing (Vaswani et al., 2017; Brown et al., 2020; Devlin et al., 2019) and computer vision (Carion et al., 2020; Dosovitskiy et al., 2021), in the RL setting they are difficult to train and are prone to overfitting (Parisotto et al., 2020). In contrast, LSTMs have long been demonstrated to be useful in RL. Although less able to capture long range dependencies due to their sequential nature, LSTMs capture recent dependencies effectively. We propose a simple but powerful architectural change: we add an LSTM layer on top of the GTrXL with an extra gated residual connection between the LSTM and GTrXL, modulated by the input to the GTrXL (see Fig 1A). Finally we also have a skip connection from the transformer input to the LSTM output.

More concretely, let $Y_t$ be the output of the encoder network at time $t$, then the additional module can be defined by the following equations (see Fig 1A, Gate(), below, has the same form as other gates internal to GTrXL):

$$X_t = \text{GTrXL}(Y_t); \quad Z_t = \text{Gate}(Y_t, X_t); \quad \text{Output}_t = \text{concatenate}(\text{LSTM}(Z_t), Y_t) \tag{4}$$

The architecture of COBERL is based on the idea that LSTMs and Transformer can be complementary. In particular, the LSTM is known for having a short contextual memory. However, by putting a transformer before the LSTM, the embeddings provided to the LSTM have already benefited from the ability of the Transformer to process long contextual dependencies, thus helping the LSTM in this respect. The second benefit works in a complementary direction. Transformers suffer from a quadratic computational complexity with respect to the sequence length. Our idea is that by letting an LSTM process some of this contextual information we can reduce the memory size of the Transformer up to the point where we see a loss in performance. This hypothesis come from studies showing that adding recurrent networks in architectures has the ability to closely emulate the behavior of non-recurrent but deeper models, but it does so with far fewer parameters (Schwarzschild et al., 2021). Having fewer parameters is an aspect of particular interest in RL, where gradients are noisy and hence training larger model is more complicated than supervised learning (McCandlish et al., 2018). Both hypotheses have been empirically confirmed by our ablations in section 4.2.

The learnt gate (eq. 4 and Appendix K), was done to give COBERL the ability to initially skip the output of the transformer stack until the weights of this are warmed-up. Once the transformer starts to output useful information the agent will be able to tune the weights in the gate to learn from them. We hypothesize that this would also help in terms of data efficiency as at the beginning of training the agent could only use the LSTM to get off the ground and start collecting relevant data to train the Transformer. Finally, the skip connection on the output of the LSTM was taken from Kapturowski et al. (2018).

Since the architecture is agnostic to the choice of RL regime we evaluate it in both on-policy and off-policy settings. For on-policy, we use V-MPO (Song et al., 2019), and for off-policy we use R2D2 (Kapturowski et al., 2018).

**R2D2 Agent**  Recurrent Replay Distributed DQN (R2D2; Kapturowski et al., 2018) demonstrates how replay and the RL learning objective can be adapted to work well for agents with recurrent architectures. Given its competitive performance on Atari-57, we implement our CoBERL architecture in the context of Recurrent Replay Distributed DQN (Kapturowski et al., 2018). We effectively replace the LSTM with our gated transformer and LSTM combination and add the contrastive representation learning loss. With R2D2 we thus leverage the benefits of distributed experience collection, storing the recurrent agent state in the replay buffer, and "burning in" a portion of the unrolled network with replayed sequences during training.

**V-MPO Agent**  Given V-MPO's strong performance on DMLab-30, in particular in conjunction with the GTrXL architecture (Parisotto et al., 2020) which is a key component of CoBERL, we use V-MPO and DMLab-30 to demonstrate CoBERL's use with on-policy algorithms. V-MPO is an on-policy adaptation of Maximum a Posteriori Policy Optimization (MPO) (Abdolmaleki et al., 2018). To avoid high variance often found in policy gradient methods, V-MPO uses a target distribution for policy updates, subject to a sample-based KL constraint, and gradients are calculated to partially move the parameters towards the target, again subject to a KL constraint. Unlike MPO, V-MPO uses a learned state-value function $V(s)$ instead of a state-action value function.

## 3  RELATED WORK

**Transformers in RL.**  The transformer architecture (Vaswani et al., 2017) has recently emerged as one of the best performing approaches in language modelling (Dai et al., 2019; Brown et al., 2020) and question answering (Dehghani et al., 2018; Yang et al., 2019). More recently it has also been successfully applied to computer vision (Dosovitskiy et al., 2021). Given the similarities of sequential data processing in language modelling and reinforcement learning, transformers have also been successfully applied to the RL domain, where as motivation for GTrXL, Parisotto et al. (2020) noted that extra gating was helpful to train transformers for RL due to the high variance of the gradients in RL relative to that of (un)supervised learning problems. In this work, we build upon GTrXL and demonstrate that, perhaps for RL: attention is not all you need, and by combining GTrXL in the right way with an LSTM, superior performance is attained. We reason that this demonstrates the advantage of both forms of memory representation: the all-to-all attention of transformers combined with the sequential processing of LSTMs. In doing so, we demonstrate that care should be taken in how LSTMs and transformers are combined and show a simple gating is most effective in our experiments. Also, unlike GTrXL, we show that using an unsupervised representation learning loss that enforces self-attention consistency is an effective way to enhance data efficiency when using transformers in RL.

**Contrastive Learning.**  Recently contrastive learning (Hadsell et al., 2006; Gutmann & Hyvärinen, 2010; Oord et al., 2018) has emerged as a very performant paradigm for unsupervised representation learning, in some cases even surpassing supervised learning (Chen et al., 2020; Caron et al., 2020; Mitrovic et al., 2021). These methods have also been leveraged in an RL setting with the hope of improving performance. Apart from MRA (Fortunato et al., 2019) mentioned above, one of the early examples of this is CURL (Srinivas et al., 2020a) which combines Q-Learning with a separate encoder used for representation learning with the InfoNCE loss from CPC (Oord et al., 2018). More recent examples use contrastive learning for predicting future latent states (Schwarzer et al., 2020; Mazoure et al., 2020), defining a policy similarity embeddings (Agarwal et al., 2021) and learning abstract representations of state-action pairs (Liu et al., 2021). The closest work to our is M-CURL (Zhu et al., 2020). Like our work, it combines mask prediction, transformers, and contrastive learning, but there are a few key differences. First, unlike M-CURL who use a separate policy network, COBERL computes Q-values based on the output of the transformer. Second, COBERL combines the transformer architecture with a learnt gate (eq. 5 in App. K to produce the input for the Q-network, while M-CURL uses the transformer as an additional embedding network (critic) for the computation of the contrastive loss. Third, while COBERL uses an extension of RELIC (Mitrovic et al., 2021) to the time domain and operates on the inputs and outputs of the transformer, M-CURL uses CPC (Oord et al., 2018) with a momentum encoder as in (Srinivas et al., 2020a) and compares encodings from the transformer with the separate momentum encoder.

## 4  EXPERIMENTS

We provide empirical evidence to show that COBERL i) improves data efficiency across a wide range of environments and tasks, and ii) needs all its components to maximise its performance. In

our experiments, we demonstrate performance on Atari57 (Bellemare et al., 2013), the DeepMind Control Suite (Tassa et al., 2018), and the DMLab-30 (Beattie et al., 2016). Recently, Dreamer V2 (Hafner et al., 2020b) has emerged as a strong model-based agent across Atari57 and DeepMind Control Suite; we therefore include it as a reference point for performance on these domains.

For all experiments, we report scores at the end of training. All results are averaged over five seeds and reported with standard error. For a more comprehensive description of the evaluation done, as well as detailed information on the distributed setup used see App. A.1. Ablations are run on 7 Atari games chosen to match the ones in the original DQN publication (Mnih et al., 2013), and on all the 30 DMLab games. The hyper-parameters of all the baselines are tuned individually to maximise performance (see App. C.5 for the detailed procedure). We use a ResNet as the encoder for CoBERL. We use Peng's $Q(\lambda)$ as our loss (Peng & Williams, 1994). To ensure that this is comparable to R2D2, we also run an R2D2 baseline with this loss. For thorough description of the architectures for all environments see App. B.

For DMLab-30 we use V-MPO (Song et al., 2019) to directly compare CoBERL with (Parisotto et al., 2020) and also demonstrate how CoBERL may be applied to both on and off-policy learning. The experiments were run using a Podracer setup (Hessel et al., 2021), details of which may be found in App. A.2. CoBERL is trained for 10 billion steps on all 30 DMLab-30 games at the same time, to mirror the exact multi-task setup presented in (Parisotto et al., 2020). Compared to (Parisotto et al., 2020) we have two differences. Firstly, all the networks run without pixel control loss (Jaderberg et al., 2016) so as not to confound our contrastive loss with the pixel control loss. Secondly all the models used a fixed set of hyperparameters with 3 random seeds, whereas in (Parisotto et al., 2020) the results were averaged across hyperparameters.

### 4.1 TESTING CoBERL ACROSS SEVERAL ENVIRONMENTS

To test the generality of our approach, we analyze the performance of our model on a wide range of environments. We show results on the Arcade Learning Environment (Bellemare et al., 2013), DeepMind Lab (Beattie et al., 2016), as well as the DeepMind Control Suite (Tassa et al., 2018). To help with comparisons, in Atari-57 and DeepMind Control we introduce an additional baseline, which we name R2D2-GTrXL. In this variant of R2D2 the LSTM is replaced by GTrXL. R2D2-GTrXL has no unsupervised learning. This way we are able to observe how GTrXL is affected by the change to an off-policy agent (R2D2), from its original V-MPO implementation in Parisotto et al. (2020). We also perform an additional ablation analysis by removing the contrastive loss from CoBERL (see Sec. 4.2.1). This baseline demonstrates the importance of contrastive learning in these domains, and we show that the combination of an LSTM and transformer is superior to either alone.

|  | > h. | Mean | Median | 25th Pct | 5th Pct |
|---|---|---|---|---|---|
| CoBERL | **49** | **1368.37% ± 34.10%** | **296.22%** | **150.56%** | **13.65%** |
| R2D2-GTrXL | 48 | 1088.90% ± 53.45% | 273.83% | 141.66% | 5.94% |
| R2D2 | 47 | 1017.18% ± 33.85% | 269.26% | 118.96% | 4.18% |
| Rainbow | 43 | 874.0% | 231.0% | 101.7% | 4.9% |
| Dreamer V2* | 37 | 631.1% | 162.0% | 76.6% | 2.5% |

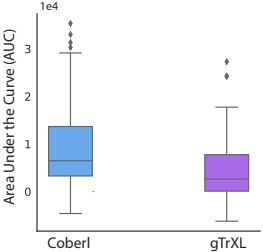

Table 1: The human normalized scores on Atari-57. > h indicates the number of tasks for which performance above average human was achieved. * indicates that it was run on 55 games with sticky actions; Pct refers to percentile.

Figure 2: Area under the curve (AUC) for Atari on all 57 levels (higher is better).

**Atari** As commonly done in literature (Mnih et al., 2015; Hessel et al., 2018; Machado et al., 2018; Hafner et al., 2020b), we measure performance on all 57 Atari games after running for 200 million frames. As detailed in App. C, we use the standard Atari frame pre-processing to obtain the 84x84 gray-scaled frames that are used as input to our agent. We do not use frame stacking.

Table 1 shows the results of all the agents where published results are available. CoBERL shows the most games above average human performance and significantly higher overall mean performance. Interestingly, the performance of R2D2-GTrXL shows that the addition of GTrXL is not sufficient to obtain the improvement in performance that CoBERL exhibits (in 4.2 we will demonstrate that both the contrastive loss and LSTM contribute to this improvement). R2D2-GTrXL also exhibits slightly

better median than CoBERL, showing that R2D2-GTrXL is indeed a powerful variant on Atari. Additionally, we observe that the difference in performance in CoBERL is higher when examining the lower percentiles. This suggests that CoBERL causes an improvement in data efficiency. To confirm this tendency we calculated the area under the curve (AUC) for the learning curves presented in Appendix F and we perform a t-test analysis on it (please see App.I for details on how AUC was calculated). Figure 2 present this analysis, and it confirms a significant difference with respect to the model used, t(114)=3.438, p=.008, with CoBERL(M=7380.23, SD=942.29) being better than GTrXL (M=5192.76, SD=942.01), thus showing that CoBERL is more data efficient.

**Control Suite** We also perform experiments on the DeepMind Control Suite (Tassa et al., 2018). While the action space in this domain is typically treated as continuous, we discretize the action space in our experiments to apply the same architecture as in Atari and DMLab-30. For more details on the number of actions for each task see App. C.4. We do not use pre-processing on the environment frames. Finally, CoBERL is trained only from pixels without state information.

| DM Suite | CoBERL | R2D2-GTrXL | R2D2 | D4PG-Pixels |
|---|---|---|---|---|
| acrobot swingup | **355.25 ± 3.82** | 265.90 ± 113.27 | 327.16 ± 5.35 | 81.7 ± 4.4 |
| fish swim | **597.66 ± 60.09** | 82.30 ± 265.07 | 345.63 ± 227.44 | 72.2 ± 3.0 |
| fish upright | **952.60 ± 5.16** | 844.13 ± 21.70 | 936.09 ± 11.58 | 405.7 ± 19.6 |
| pendulum swingup | **835.06 ± 9.38** | 775.72 ± 50.06 | 831.86 ± 61.54 | 680.9 ± 41.9 |
| swimmer swimmer6 | **419.26 ± 48.37** | 206.95 ± 58.37 | 329.61 ± 26.77 | 194.7 ± 15.9 |
| finger spin | **985.96 ± 1.69** | 983.74 ± 10.23 | 980.85 ± 0.67 | **985.7 ± 0.6** |
| reacher easy | **984.00 ± 2.76** | **982.60 ± 2.24** | **982.28 ± 9.30** | 967.4 ± 4.1 |
| cheetah run | **523.36 ± 41.38** | 105.23 ± 144.82 | 365.45 ± 50.40 | **523.8 ± 6.8** |
| walker walk | 781.16 ± 26.16 | 584.50 ± 70.28 | 687.18 ± 18.15 | **968.3 ± 1.8** |
| ball in cup catch | 979.10 ± 6.12 | 975.26 ± 1.63 | **980.54 ± 1.94** | 980.5 ± 0.5 |
| cartpole swingup | 812.01 ± 7.61 | 836.01 ± 4.37 | 816.23 ± 2.93 | **862.0 ± 1.1** |
| cartpole swingup sparse | 714.80 ± 16.18 | 743.71 ± 9.27 | **762.57 ± 6.71** | 482.0 ± 56.6 |

Table 2: Results on tasks in the DeepMind Control Suite. CoBERL, R2D2-GTrXL, R2D2, and D4PG-Pixels are trained on 100M frames. In the appendix (Table 17), we show three other approaches as reference and not as a directly comparable baseline.)

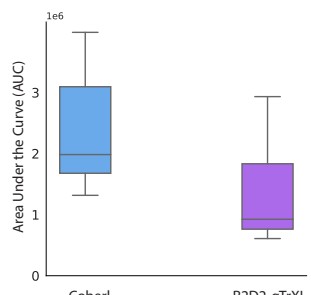

Figure 3: Area under the curve (AUC) for DmControl (higher is better).

We include six tasks popular in current literature: `ball_in_cup catch`, `cartpole swingup`, `cheetah run`, `finger spin`, `reacher easy`, and `walker walk`. Most previous work on these specific tasks has emphasized data efficiency as most are trivial to solve even with the baseline—D4PG-Pixels—in the original dataset paper (Tassa et al., 2018). We thus include 6 other tasks that are difficult to solve with D4PG-Pixels and are relatively less explored: `acrobot swingup`, `cartpole swingup_sparse`, `fish swim`, `fish upright`, `pendulum swingup`, and `swimmer swimmer6`. In Table 2 we show results on CoBERL, R2D2-gTRXL, R2D2, CURL (Srinivas et al., 2020a), Dreamer (Hafner et al., 2020a), Soft Actor Critic (Haarnoja et al., 2018) on pixels as demonstrated in (Srinivas et al., 2020a), and D4PG-Pixels (Tassa et al., 2018). CURL, DREAMER, and Pixel SAC are for reference only as they represent the state the art for low-data experiments (500K environment steps). These three are not perfectly comparable baselines; however, D4PG-Pixels is run on a comparable scale with 100 million environment steps. Because CoBERL relies on large scale distributed experience, we have a much larger number of available environment steps per gradient update. We run for 100M environment steps as with D4PG-Pixels, and we compute performance for our approaches by taking the evaluation performance of the final 10% of steps. Across the majority of tasks, CoBERL outperforms D4PG-Pixels. The increase in performance is especially apparent for the more difficult tasks. For most of the easier tasks, the performance difference between the CoBERL, R2D2-GTrXL, and R2D2 is negligible. For `ball_in_cup catch`, `cartpole swingup`, `finger spin` and `reacher easy`, even the original R2D2 agent performs on par with the D4PG-Pixels baseline. On more difficult tasks such as `fish swim`, and `swimmer swimmer6`, there is a very large, appreciable difference between CoBERL, R2D2, and R2D2-GTrXL. The combination of the LSTM and transformer specifically makes a large difference here especially compared to D4PG-Pixels. Interestingly, this architecture is also very important for situations where the R2D2-based approaches underperform. For `cheetah run` and `walker walk`, CoBERL dramatically narrows the performance gap between R2D2 and state of the art (in App. F we report learning curves for each game). Figure 3 shows that also in this domain, we see a significant AUC improvement for CoBERL when compared to R2D2-gTrXL, t(24)=1.609, p=.03.

**DMLab-30.** To test CoBERL in a challenging 3 dimensional environment we run it in DmLab-30 (Beattie et al., 2016). The agent was trained at the same time on all the 30 tasks, following

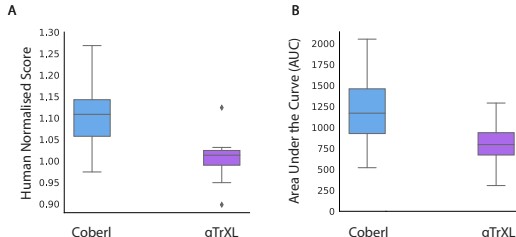

Figure 4: DMLab-30 experiments. a) Human normalised returns in DMLab-30 across all the 30 levels (higher is better). b) Area under the curve for all the 30 DMLab levels. Results are over 5 seeds and the final 5% of training.(lower) c) Area under the curve (AUC) for DMLab-30 across all the 30 levels. (higher is better)

the setup of GTrXl (Parisotto et al., 2020), which we use as our baseline. In Figure 4A we show the final results on the DMLab-30 domain. If we look at all the 30 games, COBERL reaches a substantially higher score than GTrXL (COBERL=115.47% ± 4.21%, GTrXL=101.54% ± 0.50%, t(60)=4.37, p=1.14e-5, Figure 4A). In Figure 4B we analysed data efficiency by computing the AUC and associated average statistics. In DMLab the difference between models is even more significant than Atari, t(60)=6.097, p=9.39e-09, with COBERL (M=1193.28, SD=383.29) having better average AUC than GTrXL (M=764.09, SD=224.51), hence showing that our methods scales well to more complex domains. (see Appendix F for learning curves)

## 4.2 ABLATIONS

In Sec. 2, we explained contributions that are essential to COBERL. We now disentangle the added benefit of these two separate contributions. Moreover, we run a set of ablations to understand the role of model size on the results. Ablations are run on 7 Atari games chosen to match the ones in the original DQN publication (Mnih et al., 2013), and on all the 30 DMLab games.

### 4.2.1 IMPACT OF AUXILIARY LOSSES

In Table 3 we show that our contrastive loss contributes to a significant gain in performance, both in Atari and DMLab-30, when compared to COBERL without it. Also, in challenging environments like DmLab-30, COBERL without extra loss is still superior to the relative baseline. The only case where we do not see and advantage of using the auxiliary loss is if we consider the median score on the reduced ablation set of Atari games. However in the case of the DmLab-30, where we consider a larger set of levels (7 vs. 30), there is a clear benefit of the auxiliary loss.

Moreover, Table 4 reports a comparison between our loss, SimCLR (Chen et al., 2020) and CURL (Srinivas et al., 2020a). Although simpler than both SimCLR - which in its original implementation requires handcrafted augmentations - and CURL - which requires an additional network - our contrastive method shows improved performance. These experiments where run only on Atari to reduce computational costs while still being sufficient for the analysis. We also ran one extra ablation study where we did not use masking of the input to make our setup as close as possible to (Mitrovic et al., 2021). As seen on Table 4, the column CoBERL w/o masking shows that removing the masking technique derived from the language literature makes the results substantially worse.

|  |  | COBERL | COBERL w/o aux loss | GTrXL baseline* |
|---|---|---|---|---|
| DMLab-30 | Mean | **115.47% ± 4.21%** | 105.29% ± 2.02% | 101.54% ± 0.50% |
|  | Median | **110.86%** | 104.84% | 101.35% |
| Atari | Mean | **726.34% ± 44.06%** | 490.61% ± 22.48% | 605.62% ± 47.46% |
|  | Median | 270.03% | 357.96% | **394.53%** |

Table 3: Impact of contrastive loss. Human normalized scores on Atari-57 ablation tasks and DMLab-30 tasks. * for DMLab the baseline is GTrXL trained with VMPO, for Atari the baseline is GTrXL trained with R2D2.

|  |  | COBERL | COBERL with CURL | COBERL with Sim-CLR | COBERL w/o masking |
|---|---|---|---|---|---|
| Atari | Mean | **726.34% ± 44.06%** | 608.53% ± 62.93% | 347.83% ± 46.74% | 331.05% ± 59.05% |
|  | Median | 270.03% | **272.34%** | 265.00% | 233.63% |

Table 4: Comparison with alternative auxiliary losses.

### 4.2.2 IMPACT OF ARCHITECTURAL CHANGES

Table 5 shows the effects of removing the LSTM from COBERL (column "w/o LSTM"), as well as removing the gate and its associated skip connection (column "w/o Gate"). In both cases COBERL performs substantially worse showing that both components are needed. Finally, we also experimented with substituting the learned gate with either a sum or a concatenation. The results, presented in Appendix D, show that in most occasions these alternatives decrease performance, but not as substantially as removing the LSTM, gate or skip connections. Our hypothesis is that the learned gate gives more flexibility in complex environments, we leave it open for future work to explore this.

|  |  | COBERL | w/o LSTM | w/o Gate |
|---|---|---|---|---|
| DMLab-30 | Mean | **115.47**% ± **4.21**% | 96.29% ± 3.39 | 86.42% ± 7.25% |
|  | Median | **110.86**% | 99.83% | 95.21% |
| Atari | Mean | **726.34**% ± **44.06** | 504.42% ± 48.51% | 528.61% ± 73.21% |
|  | Median | 270.03% | 249.62% | **283.16**% |

Table 5: Impact of Architectural changes. Human normalized scores on Atari ablation tasks and DMLab-30.

### 4.2.3 IMPACT OF NUMBER OF PARAMETERS

Table 6 compares the models in terms of the number of parameters. For Atari, the number of parameters added by COBERL over the R2D2(GTrXL) baseline is very limited; however, COBERL still produces a significant gain in performance. We also tried to move the LSTM before the transformer module (column "COBERL with LSTM before"). In this case the representations for the contrastive loss were taken from before the LSTM. Interestingly, this setting performs worse, despite having the same number of parameters as COBERL. This goes in the direction of our hypothesis that having the LSTM after the Transformer could allow the former to exploit the extra context provided by the latter. For DMLab-30, it is worth noting that COBERL has a memory size of 256, whereas GTrXL has a memory of size 512 resulting in substantially fewer parameters. Nevertheless, the discrepancies between models are even more pronounced, even though the number of parameters is either exactly the same ("COBERL with LSTM before") or higher (GTrXL). This ablation is of particular interest as it shows that the results are driven by the particular architectural choice rather than the added parameters. Also it helps supporting our idea that the LSTM, by processing a certain amount of contextual information, allow for a reduced memory size on the Transformer, without a loss in performance and with a reduce computation complexity (given fewer parameters).

|  |  | COBERL | GTrXL* | COBERL with LSTM before | R2D2 LSTM |
|---|---|---|---|---|---|
| DMLab-30 | Mean | **115.47**% ± **4.21**% | 101.54% ± 0.50% | 101.29% ± 1.80% | N/A |
|  | Median | **110.86**% | 103.82% | 100.03% | N/A |
|  | Num. Params. | 47M | 66M | 47M | N/A |
| Atari | Mean | **726.34**% ± **44.06** | 605.62% ± 47.46% | 604.36% ± 78.60% | 362.38% ± 20.28% |
|  | Median | 270.03% | **394.53**% | 296.72% | 202.71% |
|  | Num. Params. | 46M | 42M | 46M | 18M |

Table 6: Effect of number of parameters. Human normalized scores on Atari-57 ablation tasks and DMLab-30 tasks. *for DMLab-30 the baseline is GTrXL trained with VMPO with a memory size of 512, for Atari the baseline is GTrXL trained with R2D2 with a memory size of 64.

## 5 CONCLUSIONS

We proposed a novel RL agent, Contrastive BERT for RL (COBERL), which introduces a new contrastive representation learning loss that enables the agent to efficiently learn consistent representations. This, paired with an improved architecture, resulted in better data efficiency on a varied set of environments and tasks. Moreover, through an extensive set of ablation experiments we confirmed that all COBERL components are necessary to achieve the performance of the final agent. Critically, COBERL is fully end-to-end as it does not require an extra encoder network (vs CURL or RELIC), or data augmentations (vs SIMCLR and RELIC) and it has fewer parameters then gTrXL. To conclude, we have shown that our auxiliary loss and architecture provide an effective and general means to efficiently train large attentional models in RL. (For extra conclusions and limitations see App.J)

# 6 OPTIONAL REPRODUCIBILITY STATEMENT

To help with reproducibility we have included extensive details for

- Setup details in Appendix A
- Architecture description in Appendix B
- Hyper-parameters chosen in Appendix C

We also report the pseudo-code for the algorithm and the auxiliary loss in Appendix H

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

# A SETUP DETAILS

## A.1 R2D2 DISTRIBUTED SYSTEM SETUP

Following R2D2, the distributed system consists of several parts: actors, a replay buffer, a learner, and an evaluator. Additionally, we introduce a centralized batched inference process to make more efficient use of actor resources.

**Actors**: We use $512$ processes to interact with independent copies of the environment, called actors. They send the following information to a central batch inference process:

- $x_t$: the observation at time $t$.
- $r_{t-1}$: the reward at the previous time, initialized with $r_{-1} = 0$.
- $a_{t-1}$: the action at the previous time, $a_{-1}$ is initialized to $0$.
- $h_{t-1}$: recurrent state at the previous time, is initialized with $h_{-1} = 0$.

They block until they receive $Q(x_t, a; \theta)$. The $l$-th actor picks $a_t$ using an $\epsilon_l$-greedy policy. As R2D2, the value of $\epsilon_l$ is computed following:

$$\epsilon_l = \epsilon^{1+\alpha \frac{l}{L-1}}$$

where $\epsilon = 0.4$ and $\alpha = 7$. After that is computed, the actors send the experienced transition information to the replay buffer.

**Batch inference process**: This central batch inference process receives the inputs mentioned above from all actors. This process has the same architecture as the learner with weights that are fetched from the learner every $0.5$ seconds. The process blocks until a sufficient amount of actors have sent inputs, forming a batch. We use a batch size of $64$ in our experiments. After a batch is formed, the neural network of the agent is run to compute $Q(x_t, a, \theta)$ for the whole batch, and these values are sent to their corresponding actors.

**Replay buffer:** it stores fixed-length sequences of *transitions* $T = (\omega_s)_{s=t}^{t+L-1}$ along with their priorities $p_T$, where $L$ is the trace length we use. A transition is of the form $\omega_s = (r_{s-1}, a_{s-1}, h_{s-1}, x_s, a_s, h_s, r_s, x_{s+1})$. Concretely, this consists of the following elements:

- $r_{s-1}$: reward at the previous time.
- $a_{s-1}$: action done by the agent at the previous time.
- $h_{s-1}$: recurrent state (in our case hidden state of the LSTM) at the previous time.
- $x_s$: observation provided by the environment at the current time.
- $a_s$: action done by the agent at the current time.
- $h_s$: recurrent state (in our case hidden state of the LSTM) at the current time.
- $r_s$: reward at the current time.
- $x_{s+1}$: observation provided by the environment at the next time.

The sequences never cross episode boundaries and they are stored into the buffer in an overlapping fashion, by an amount which we call the *replay period*. Finally, concerning the priorities, we followed the same prioritization scheme proposed by Kapturowski et al. (2018) using a mixture of max and mean of the TD-errors in the sequence with priority exponent $\eta = 0.9$.

**Evaluator**: the evaluator shares the same network architecture as the learner, with weights that are fetched from the learner every episode. Unlike the actors, the experience produced by the evaluator is not sent to the replay buffer. The evaluator acts in the same way as the actors, except that all the computation is done within the single CPU process instead of delegating inference to the batch inference process. At the end of $5$ episodes the results of those $5$ episodes are average and reported. In this paper we report the average performance provided by such reports over the last $5\%$ frames (for example, on Atari this is the average of all the performance reports obtained when the total frames consumed by actors is between 190M and 200M frames).

**Learner**: The learner contains two identical networks called the online and target networks with different weights $\theta$ and $\theta'$ respectively (Mnih et al., 2015). The target network's weights $\theta'$ are updated to $\theta$ every $400$ optimization steps. $\theta$ is updated by executing the following sequence of instructions:

- First, the learner samples a batch of size $64$ (batch size) of fixed-length sequences of transitions from the replay buffer, with each transition being of length $L$: $T_i = (\omega_s^i)_{s=t}^{t+L-1}$.
- Then, a forward pass is done on the online network and the target with inputs $(x_s^i, r_{s-1}^i, a_{s-1}^i, h_{s-1}^i)_{s=t}^{t+H}$ in order to obtain the state-action values $\{(Q(x_s^i, a; \theta), Q(x_s^i, a; \theta'))\}$.
- With $\{(Q(x_s^i, a; \theta), Q(x_s^i, a; \theta'))\}$, the $Q(\lambda)$ loss is computed.
- The online network is used again to compute the auxiliary contrastive loss.
- Both losses are summed (with by weighting the auxiliary loss by $0.1$ as described in C), and optimized with an Adam optimizer.
- Finally, the priorities are computed for the sampled sequence of transitions and updated in the replay buffer.

### A.2 V-MPO DISTRIBUTED SETUP

For on-policy training, we used a Podracer setup similar to (Hessel et al., 2021) for fast usage of experience from actors by learners.

**TPU learning and acting**: As in the Sebulba setup of (Hessel et al., 2021), acting and learning network computations were co-located on a set of TPU chips, split into a ratio of 3 cores used for learning for every 1 core used for inference. This ratio then scales with the total number of chips used.

**Environment execution**: Due to the size of the recurrent states used by COBERL and stored on the host CPU, it was not possible to execute the environments locally. To proceed we used 64 remote environment servers which serve only to step multiple copies of the environment. 1024 concurrent episodes were processed to balance frames per second, latency between acting and learning, and memory usage of the agent states on the host CPUs.

### A.3 COMPUTATION USED

**R2D2** We train the agent with a single TPU v2-based learner, performing approximately 5 network updates per second (each update on a mini-batch of $64$ sequences of length $80$ for Atari and $120$ for Control). We use $512$ actors, using $4$ actors per CPU core, with each one performing $\sim 64$ environment steps per second on Atari. Finally for the batch inference process a TPU v2, which allows all actors to achieve the speed we have described. In particular, we used 8 TPU cores for learning and 2 for inference.

**V-MPO** We train the agent with 4 hosts each with 8 TPU v2 cores. Each of the 8 cores per host was split into 6 for learning and 2 for inference. We separately used 64 remote CPU environment servers to step 1024 concurrent environment episodes using the actions returned from inference. The learner updates were made up of a mini-batch of 120 sequences, each of length 95 frames. This setup enabled 4.6 network updates per second, or 53.4k frames per second.

### A.4 COMPLEXITY ANALYSIS

As stated, the agent consists of layers of convolutions, transformer layers, and linear layers. Therefore the complexity is $max\{O(n^2 \cdot d), O(k \cdot n \cdot d^2)\}$, where $k$ is the kernel size in the case of convolutions, $n$ is the size of trajectories, and $d$ is the size of hidden layers.

## B    ARCHITECTURE DESCRIPTION

### B.1    ENCODER

As shown in Fig. 1, observations $O_i$ are encoded using an encoder. In this work, the encoder we have used is a ResNet-47 encoder. Those 47 layers are divided in 4 groups which have the following characteristics:

- An initial stride-2 convolution with filter size 3x3 ($1 \cdot 4$ layers).
- Number of residual bottleneck blocks (in order): $(2, 4, 6, 2)$. Each block has 3 convolutional layers with ReLU activations, with filter sizes 1x1, 3x3, and 1x1 respectively ($(2+4+6+2)\cdot 3$ layers).
- Number of channels for the last convolution in each block: $(64, 128, 256, 512)$.
- Number of channels for the non-last convolutions in each block: $(16, 32, 64, 128)$.
- Group norm is applied after each group, with a group size of $8$.

After this observation encoding step, a final 2-layer MLP with ReLU activations of sizes $(512, 448)$ is applied. The previous reward and one-hot encoded action are concatenated and projected with a linear layer into a $64$-dimensional vector. This $64$-dimensional vector is concatenated with the $448$-dimensional encoded input to have a final $512$-dimensional output.

### B.2    TRANSFORMER

As described in Section 2, the output of the encoder is fed to a Gated Transformer XL. For Atari and Control, the transformer has the following characteristics:

- Number of layers: $8$.
- Memory size: $64$.
- Hidden dimensions: $512$.
- Number of heads: $8$.
- Attention size: $64$.
- Output size: $512$.
- Activation function: GeLU.

For DmLab the transformer has the following characteristics:

- Number of layers: $12$.
- Memory size: $256$ for COBERL and $512$ for gTrXL.
- Hidden dimensions: $128$.
- Number of heads: $4$.
- Attention size: $64$.
- Output size: $512$.
- Activation function: ReLU.

the GTrXL baseline is identical, but with a Memory size of $512$.

### B.3    LSTM AND VALUE HEAD

For both R2D2 and V-MPO the outputs of the transformer and encoder are passed through a GRU transform to obtain a $512$-dimensional vector. After that, an LSTM with $512$ hidden units is applied. The the value function is estimated differently depending on the RL algorithm used.

**R2D2**  Following the LSTM, a Linear layer of size 512 is used, followed by a ReLU activation. Finally, to compute the Q values from that 512 vector a dueling head is used, as in Kapturowski et al. (2018), a dueling head is is used, which requires a linear projection to the number of actions of the task, and another projection to a unidimensional vector.

**V-MPO**  Following the LSTM, a 2 layer MLP with size 512 and 30 (i.e. the number of levels in DMLab) is used. In the MLP we use ReLU activation. As we are interested in the multi-task setting where a single agent learns a large number of tasks with differing reward scales, we used PopArt (van Hasselt et al., 2016) for the value function estimation (see Table. 13 for details).

### B.4 CRITIC FUNCTION

For DmLab-30 (V-MPO), we used a 2 layer MLP with hidden sizes 512 and 128. For Atari and Control Suite (R2D2) we used a single layer of size 512.

## C  HYPERPARAMETERS

For the experiments in Atari57 and the DeepMind Control suite, COBERL uses the R2D2 distributed setup. We use $512$ actors for all our experiments. We do not constrain the amount of replay done for each experience trajectory that actors deposit in the buffer. However, we have found empirical replay frequency per data point to be close among all our experiments (with an expected value of $1.5$ samples per data point). We use a separate evaluator process that shares weights with our learner in order to measure the performance of our agents. We report scores at the end of training. The hyperparameters and architecture we choose for these two domains are the same with two exceptions: i) we use a shorter trace length for Atari ($80$ instead of $120$) as the environment does not require a long context to inform decisions, and ii) we use a squashing function on Atari and the Control Suite to transform our $Q$ values (as done in (Kapturowski et al., 2018)) since reward structures vary highly in magnitude between tasks.

### C.1  ATARI AND DMLAB PRE-PROCESSING

We use the commonly used input pre-processing on Atari and DMLab frames, shown on Tab. 8. One difference with the original work of Mnih et al. (2015), is that we do not use frame stacking, as we rely on our memory systems to be able to integrate information from the past, as done in Kapturowski et al. (2018). ALE is publicly available at `https://github.com/mgbellemare/Arcade-Learning-Environment`.

| Hyperparameter | Value |
|---|---|
| Max episode length | $30\ min$ |
| Num. action repeats | 4 |
| Num. stacked frames | 1 |
| Zero discount on life loss | $false$ |
| Random noops range | 30 |
| Sticky actions | $false$ |
| Frames max pooled | 3 and 4 |
| Grayscaled/RGB | Grayscaled |
| Action set | Full |

Table 7: Atari pre-processing hyperparameters.

### C.2  CONTROL SUITE PRE-PROCESSING

As mentioned in 4, we use no pre-processing on the frames received from the control environment.

### C.3  DMLAB PRE-PROCESSING

| Hyperparameter | Value |
|---|---|
| Num. action repeats | 4 |
| Num. stacked frames | 1 |
| Grayscaled/RGB | RGB |
| Image width | 96 |
| Image height | 72 |
| Action set | as in Parisotto et al. (2020) |

Table 8: DmLab pre-processing hyperparameters.

### C.4  CONTROL ENVIRONMENT DISCRETIZATION

As mentioned, we discretize the space assigning two possibilities (1 and -1) to each dimension and taking the Cartesian product of all dimensions, which results in $2^n$ possible actions. For the `cartpole` tasks, we take a diagonal approach, utilizing each unit vector in the action space and

then dividing each unit vector into 5 possibilities with the non-zero coordinate ranging from -1 to 1. The amount of actions this results in is outlined on Tab. 9.

| Task | Action space size | Total amount of actions |
|---|---|---|
| Acrobot | 1 | 2 |
| Cartpole | 1 | 5 |
| Cup | 2 | 4 |
| Cheetah | 6 | 64 |
| Finger | 2 | 4 |
| Fish | 5 | 32 |
| Pendulum | 1 | 2 |
| Reacher | 2 | 4 |
| Swimmer | 5 | 32 |
| Walker | 6 | 64 |

Table 9: Control discretization action spaces.

## C.5 HYPERPARAMETERS USED

We list all hyperparameters used here for completeness.

We started by optimizing the hyperparameters of GTrXL highlighted in bold in Tab 10 by doing a sweep over 10 Atari games: Seaquest, Qbert, Frostbite, Ms Pacman, Space Invaders, Gravitar, Solaris, Hero, Venture, Montezuma Revenge. Following this, the hyperparameters were kept fixed throughout all the experiments.

Table 10 reports all the hyperparameters of the R2D2 experiments, both the fixed ones and the ones with the ranges over which we did the sweep. The fixed hyper-parameters were taken from Kapturowski et al. (2018). We then choose a set of hyper-paramters (both for the architecture and the algorithm) to sweep over to maximise the performance of gTrXL in this off-policy setting, given that there was not prior literature on this. The hyper-paramters over which we did the sweep for the algorithm were chose in accordance to Kapturowski et al. (2018) and the related sweep. And for the architecture hyper-parameters we based our choice on Parisotto et al. (2020). We believe that in this way we ensure to have a properly tuned baseline that enforces a fair comparison. Table 11 reports the chosen hyperparameters that we found to optimize the performance of GTrXL on the 10 Atari games. We then moved to CoBERL. CoBERL, in comparison to the baseline GTrXL has two extra hyperparamters: 'Contrastive loss weight' and 'Contrastive loss mask rate'. The former was tuned, whereas the latter we kept equal to 0.15 as done in [11]. Consequently, we re-ran the same procedure as before, but we fixed all the previous hyperparameters optimized for GTrXL (see Tab. 11 and we perform a grid search over the same 10 Atari games to find the value of 'Contrastive loss weight' that maximized performance. The values over which we did the search are 0.01, 0.1 and 1. We ended-up picking 1, although the difference between 0.1 and 1 was minimal. Table 12 reports the 2 extra parameters used for COBERL.

For DmLAB we optimized the hyperparameters of GTrXL on all the 30 games. Table 13 reports both the fixed ones and the ones with the ranges over which. The fixed hyper-parameters were taken directly from Parisotto et al. (2020) and we sweep over ''Epsilon Alpha'', "Target Update Period" and 'Memory size' to make sure we maximised performance of this baseline. Table 14 reports the chosen hyperparameters that we found to optimize the performance of GTrXL on DmLAB. We then moved to CoBERL. Again, by keeping fixed all the previous hyperparameters optimized for GTrXL we perform a grid search over all the 30 games to find the value of 'Contrastive loss weight' that maximized performance. The values over which we did the search were 0.1 and 1. We did not find any significant difference between the two values, so we left it equal to 1 such that the two losses would have the same effect. Table 15 reports the 2 extra parameters used for CoBERL and the reduced memory size, in accordance with our hypothesis that the LSTM on top of Transformer would help reducing the size of the memory especially in last.

| Hyperparameter | Value |
|---|---|
| Optimizer | Adam |
| Learning rate | $\{\mathbf{0.0001},\ \mathbf{0.0003}\}$ |
| Q's $\lambda$ | $\{\mathbf{0.8},\ \mathbf{0.9}\}$ |
| Adam epsilon | $10^{-7}$ |
| Adam beta1 | 0.9 |
| Adam beta2 | 0.999 |
| Adam clip norm | 40 |
| Q-value transform (non-DMLab) | $h(x) = sign(x)(\sqrt{|x| + 1} - 1) + \epsilon x$ |
| Q-value transform (DMLab) | $h(x) = x$ |
| Discount factor | 0.997 |
| Batch size | 32 |
| Trace length (Atari) | 80 |
| Trace length (non-Atari) | 120 |
| Replay period (Atari) | 40 |
| Replay period (non-Atari) | 60 |
| Replay capacity | 80000 sequences |
| Replay priority exponent | 0.9 |
| Importance sampling exponent | 0.6 |
| Minimum sequences to start replay | 5000 |
| Target Q-network update period | 400 |
| Evaluation $\epsilon$ | 0.01 |
| Target $\epsilon$ | 0.01 |
| Number of layers | $\{\mathbf{6}, \mathbf{8}, \mathbf{12}\}$ |
| Memory size | $\{\mathbf{64}, \mathbf{128}\}$ |
| Number of heads | $\{\mathbf{4}, \mathbf{8}\}$ |
| Attention size | $\{\mathbf{64}, \mathbf{128}\}$ |

Table 10: GTrXL Hyperparameters used in all the R2D2 experiments with range of sweep.

| Hyperparameter | Value |
|---|---|
| Optimizer | Adam |
| Learning rate | **0.0003** |
| Q's $\lambda$ | **0.8** |
| Adam epsilon | $10^{-7}$ |
| Adam beta1 | 0.9 |
| Adam beta2 | 0.999 |
| Adam clip norm | 40 |
| Q-value transform (non-DMLab) | $h(x) = sign(x)(\sqrt{|x| + 1} - 1) + \epsilon x$ |
| Q-value transform (DMLab) | $h(x) = x$ |
| Discount factor | 0.997 |
| Batch size | 32 |
| Trace length (Atari) | 80 |
| Trace length (non-Atari) | 120 |
| Replay period (Atari) | 40 |
| Replay period (non-Atari) | 60 |
| Replay capacity | 80000 sequences |
| Replay priority exponent | 0.9 |
| Importance sampling exponent | 0.6 |
| Minimum sequences to start replay | 5000 |
| Target Q-network update period | 400 |
| Evaluation $\epsilon$ | 0.01 |
| Target $\epsilon$ | 0.01 |
| Number of layers | **8** |
| Memory size | **64** |
| Number of heads | **8** |
| Attention size | **64** |

Table 11: GTrXL Hyperparameters choosen for all the R2D2 experiments.

| Hyperparameter | Value |
|---|---|
| Contrastive loss weight | 1.0 |
| Contrastive loss mask rate | 0.15 |

Table 12: Extra hyperparameters for CoBERL for the R2D2 experiments

| Hyperparameter | Value |
|---|---|
| Batch Size | 120 |
| Unroll Length | 95 |
| Discount | 0.99 |
| Target Update Period | $\{\mathbf{10},\ \mathbf{20},\ \mathbf{50}\}$ |
| Action Repeat | 4 |
| Initial $\eta$ | 1.0 |
| Initial $\alpha$ | 5.0 |
| $\epsilon_\eta$ | 0.1 |
| $\epsilon_\alpha$ | $\{\mathbf{0.001},\ \mathbf{0.002}\}$ |
| Popart Step Size | 0.001 |
| Memory size | $\{\mathbf{256}, \mathbf{512}\}$ |

Table 13: GTrXL Hyperparameters used in all the VMPO experiments with range of sweep.

| Hyperparameter | Value |
|---|---|
| Batch Size | 120 |
| Unroll Length | 95 |
| Discount | 0.99 |
| Target Update Period | 50 |
| Action Repeat | 4 |
| Initial $\eta$ | 1.0 |
| Initial $\alpha$ | 5.0 |
| $\epsilon_\eta$ | 0.1 |
| $\epsilon_\alpha$ | 0.002 |
| Popart Step Size | 0.001 |
| Memory size | $\mathbf{512}$ |

Table 14: GTrXL Hyperparameters choosen for all the VMPO experiments.

| Hyperparameter | Value |
|---|---|
| Contrastive loss weight | 1.0 |
| Contrastive loss mask rate | 0.15 |
| Memory size | 256 |

Table 15: Extra hyperparameters for CoBERL for the VMPO experiments

# D ADDITIONAL ABLATIONS

Table 16 shows the results of several gating mechanisms that we have investigated. As we can observe the GRU gate is a clear improvement especially on DMLab, only being harmful in median on the reduced ablation set of Atari games.

|  |  | COBERL | 'Sum' gate | 'Concat' gate | w/o Gate |
|---|---|---|---|---|---|
| DMLab | Mean | **115.47**% ± **4.21**% | 108.92% ± 4.56% | 105.93% ± 4.82% | 86.42% ± 7.25% |
| | Median | **110.86**% | 108.12% | 107.82% | 95.21% |
| Atari | Mean | **726.34**% ± **44.06**% | 548.66%±11.16% | 653.20%±59.13% | 591.33%±91.25% |
| | Median | 270.03% | **437.85**% | 325.96% | 320.09% |

Table 16: Gate ablations. Human normalized scores on Atari-57 ablation tasks and DMLab tasks. For the mean we include standard error over seeds.

| DM Suite | COBERL | R2D2-GTrXL | R2D2 | D4PG-Pixels | CURL | Dreamer | Pixel SAC |
|---|---|---|---|---|---|---|---|
| acrobot swingup | **355.25** ± **3.82** | 265.90 ± 113.27 | 327.16 ± 5.35 | 81.7 ± 4.4 | - | - | - |
| fish swim | **597.66** ± **60.09** | 82.30 ± 265.07 | 345.63 ± 227.44 | 72.2 ± 3.0 | - | - | - |
| fish upright | **952.60** ± **5.16** | 844.13 ± 21.70 | 936.09 ± 11.58 | 405.7 ± 19.6 | - | - | - |
| pendulum swingup | **835.06** ± **9.38** | 775.72 ± 50.06 | 831.86 ± 61.54 | 680.9 ± 41.9 | - | - | - |
| swimmer swimmer6 | **419.26** ± **48.37** | 206.95 ± 58.37 | 329.61 ± 26.77 | 194.7 ± 15.9 | - | - | - |
| finger spin | **985.96** ± **1.69** | 983.74 ± 10.2 | 980.85 ± 0.67 | **985.7** ± **0.6** | 926 ± 45 | 796 ± 183 | 179 ± 166 |
| reacher easy | **984.00** ± **2.76** | **982.60** ± **2.24** | **982.28** ± **9.30** | 967.4 ± 4.1 | 929 ± 44 | 793 ± 164 | 145 ± 30 |
| cheetah run | **523.36** ± **41.38** | 105.23 ± 144.82 | 365.45 ± 50.40 | **523.8** ± **6.8** | 518 ± 28 | **570** ± **253** | 197 ± 15 |
| walker walk | 781.16 ± 26.16 | 584.50 ± 70.28 | 687.18 ± 18.15 | **968.3** ± **1.8** | 902 ± 43 | 897 ± 49 | 42 ± 12 |
| ball in cup catch | 979.10 ± 6.12 | 975.26 ± 1.63 | **980.54** ± **1.94** | 980.5 ± 0.5 | 959 ± 27 | 879 ± 87 | 312 ± 63 |
| cartpole swingup | 812.01 ± 7.61 | 836.01 ± 4.37 | 816.23 ± 2.93 | **862.0** ± **1.1** | 841 ± 45 | 762 ± 27 | 419 ± 40 |
| cartpole swingup sparse | 714.80 ± 16.18 | 743.71 ± 9.27 | **762.57** ± **6.71** | 482.0 ± 56.6 | - | - | - |

Table 17: Results on tasks in the DeepMind Control Suite. CoBERL, R2D2-GTrXL, R2D2, and D4PG-Pixels are trained on 100M frames, while CURL, Dreamer, and Pixel SAC are trained on 500k frames. We show these three other approaches as reference and not as a directly comparable baseline.

# E  GAME SCORES

| Atari (ablation games) | CoBERL | R2D2-gTrXL | R2D2 | CoBERL-auxiliary loss |
|---|---|---|---|---|
| beam rider | 29601.17 ± 11973.77 | **65709.85 ± 29911.83** | 33938.58 ± 11340.33 | 43029.32 ± 47122.88 |
| enduro | 2323.66 ± 31.42 | 2247.76 ± 139.16 | **2338.98 ± 13.23** | 2250.01 ± 69.20 |
| breakout | 409.87 ± 11.33 | 379.23 ± 32.37 | 363.43 ± 98.32 | **420.72 ± 9.86** |
| pong | 21.00 ± 0.00 | 20.95 ± 0.09 | **21.00 ± 0.00** | 21.00 ± 0.00 |
| qbert | 34000.79 ± 6074.45 | 22444.14 ± 4270.57 | 25375.24 ± 7451.33 | **47741.30 ± 5288.90** |
| seaquest | 164587.95 ± 122633.80 | **310745.77 ± 67781.42** | 83490.38 ± 121434.46 | 174867.68 ± 123876.30 |
| space invaders | **37497.11 ± 9973.19** | 19113.28 ± 8512.98 | 4704.35 ± 2763.78 | 8783.28 ± 6450.09 |

| Atari (ablation games) | CoBERL | CoBERL with LSTM before | CoBERL w/o LSTM |
|---|---|---|---|
| beam rider | 29601.17 ± 11973.77 | 35188.16 ± 23357.60 | **62234.03 ± 12764.95** |
| enduro | **2323.66 ± 31.42** | 2321.02 ± 43.05 | 2286.74 ± 112.37 |
| breakout | 409.87 ± 11.33 | **418.13 ± 6.77** | 411.68 ± 13.62 |
| pong | **21.00 ± 0.00** | 21.00 ± 0.00 | 21.00 ± 0.00 |
| qbert | **34000.79 ± 6074.45** | 30055.59 ± 9892.69 | 31198.54 ± 5557.65 |
| seaquest | 164587.95 ± 122633.80 | 172136.83 ± 119874.67 | **192161.44 ± 94730.21** |
| space invaders | **37497.11 ± 9973.19** | 23746.32 ± 18961.05 | 10169.22 ± 12746.13 |

| Atari (ablation games) | CoBERL | No Skip Conn. | Sum gate | Concat |
|---|---|---|---|---|
| beam rider | 29601.17 ± 11973.77 | 52498.32 ± 2934.21 | **72882.42 ± 21239.63** | 51371.38 ± 12560.94 |
| enduro | **2323.66 ± 31.42** | 2168.37 ± 362.99 | 2247.23 ± 82.22 | 2288.13 ± 58.59 |
| breakout | **409.87 ± 11.33** | 308.68 ± 141.28 | 403.36 ± 53.69 | 357.52 ± 72.63 |
| pong | 21.00 ± 0.00 | 20.88 ± 0.18 | **21.00 ± 0.00** | 21.00 ± 0.00 |
| qbert | 34000.79 ± 6074.45 | 27660.71 ± 6830.06 | 33807.00 ± 4294.93 | **43487.35 ± 14518.46** |
| seaquest | 164587.95 ± 122633.80 | 118960.81 ± 112462.18 | 294976.55 ± 41827.55 | **399817.75 ± 36729.78** |
| space invaders | **37497.11 ± 9973.19** | 22347.29 ± 15281.67 | 10387.59 ± 2858.63 | 20933.98 ± 13072.95 |

| Atari (ablation games) | CoBERL with CURL | CoBERL with SimCLR |
|---|---|---|
| beam rider | $25605.45 \pm 12772.65$ | $\mathbf{36364.92 \pm 27431.91}$ |
| enduro | $2280.38 \pm 152.75$ | $\mathbf{2343.53 \pm 5.10}$ |
| breakout | $337.39 \pm 53.97$ | $\mathbf{397.40 \pm 40.07}$ |
| pong | $19.99 \pm 2.02$ | $\mathbf{20.94 \pm 0.11}$ |
| qbert | $15935.73 \pm 580.68$ | $\mathbf{24491.28 \pm 14491.24}$ |
| seaquest | $163715.19 \pm 80515.52$ | $\mathbf{273070.09 \pm 124529.37}$ |
| space invaders | $14038.27 \pm 16158.41$ | $\mathbf{22120.53 \pm 17354.48}$ |

| Atari-57 | CoBERL | R2D2-gTrXL | R2D2 |
|---|---|---|---|
| alien | **10728.64 ± 6771.87** | 9593.30 ± 3226.26 | 8960.97 ± 4233.52 |
| amidar | **3089.76 ± 1028.80** | 2888.93 ± 1324.27 | 1980.60 ± 181.03 |
| assault | 10055.81 ± 7856.45 | 7569.86 ± 4705.86 | **18601.87 ± 6948.52** |
| asterix | 732004.08 ± 342695.21 | 535334.42 ± 400048.35 | **777461.28 ± 251661.19** |
| asteroids | **138026.82 ± 63177.27** | 108693.26 ± 22302.14 | 56238.62 ± 44595.74 |
| atlantis | **1062735.42 ± 47026.77** | 1005098.28 ± 36831.12 | 994240.79 ± 42158.91 |
| bank heist | 1126.69 ± 620.85 | **1131.62 ± 272.21** | 1110.12 ± 442.89 |
| battle zone | 92092.43 ± 39167.40 | 77051.40 ± 45255.27 | **94903.82 ± 20522.95** |
| beam rider | 29601.17 ± 11973.77 | **65709.85 ± 29911.83** | 33938.58 ± 11340.33 |
| berzerk | 1288.99 ± 615.45 | 594.60 ± 117.78 | **1314.18 ± 444.67** |
| bowling | **160.94 ± 53.41** | 49.23 ± 47.13 | 86.34 ± 27.65 |
| boxing | 69.40 ± 51.78 | **100.00 ± 0.00** | 99.28 ± 0.79 |
| breakout | **409.87 ± 11.33** | 379.23 ± 32.37 | 363.43 ± 98.32 |
| centipede | 62816.39 ± 13516.24 | **98575.88 ± 43443.70** | 75733.83 ± 19166.94 |
| chopper command | **614750.54 ± 283486.50** | 53852.34 ± 53906.17 | 26147.63 ± 14180.38 |
| crazy climber | **136884.73 ± 50591.49** | 117777.36 ± 18382.23 | 134610.46 ± 35360.45 |
| defender | 347347.03 ± 90741.03 | 304551.03 ± 75288.86 | **383931.18 ± 151219.51** |
| demon attack | **135904.86 ± 12721.85** | 115010.85 ± 26125.20 | 96371.00 ± 51049.84 |
| double dunk | **24.00 ± 0.00** | 21.25 ± 2.76 | 23.77 ± 0.33 |
| enduro | 2323.66 ± 31.42 | 2247.76 ± 139.16 | **2338.98 ± 13.23** |
| fishing derby | 45.74 ± 9.05 | 39.83 ± 4.85 | **47.47 ± 8.12** |
| freeway | **34.00 ± 0.00** | 34.00 ± 0.00 | 33.67 ± 0.42 |
| frostbite | **7877.53 ± 1919.45** | 6758.04 ± 2227.07 | 5850.66 ± 1443.76 |
| gopher | 98169.07 ± 12126.03 | 84697.17 ± 26263.93 | **106325.84 ± 17858.36** |
| gravitar | **6219.65 ± 484.85** | 5181.25 ± 1615.50 | 5186.02 ± 808.22 |
| hero | **20487.23 ± 288.81** | 16544.03 ± 3106.35 | 16764.65 ± 2575.03 |
| ice hockey | **24.64 ± 19.00** | 9.94 ± 12.69 | 21.38 ± 15.82 |
| jamesbond | 5589.27 ± 4360.99 | **7149.04 ± 3244.40** | 5872.92 ± 3432.57 |
| kangaroo | 11703.01 ± 2892.01 | 11760.61 ± 3657.70 | **13223.97 ± 2123.85** |
| krull | 27738.24 ± 19282.30 | **42886.34 ± 35329.57** | 19821.18 ± 17250.91 |
| kung fu master | **128429.78 ± 12649.00** | 64979.95 ± 25384.52 | 99741.61 ± 14185.70 |
| montezuma revenge | **502.27 ± 998.87** | 0.00 ± 0.00 | 240.00 ± 195.96 |
| ms pacman | **10567.67 ± 3692.12** | 9905.93 ± 884.03 | 9576.99 ± 2286.69 |
| name this game | 26311.25 ± 6607.53 | **26669.02 ± 3614.10** | 23889.30 ± 2608.30 |
| phoenix | **424708.26 ± 173336.21** | 283179.23 ± 94106.97 | 138558.03 ± 154881.71 |
| pitfall | **0.00 ± 0.00** | 0.00 ± 0.00 | 0.00 ± 0.00 |
| pong | 21.00 ± 0.00 | 20.95 ± 0.09 | **21.00 ± 0.00** |
| private eye | 12034.62 ± 5963.85 | **12690.27 ± 4372.38** | 3065.63 ± 5867.30 |
| qbert | **34000.79 ± 6074.45** | 22444.14 ± 4270.57 | 25375.24 ± 7451.33 |
| riverraid | 22623.35 ± 3512.77 | 23692.95 ± 2100.11 | **28147.78 ± 678.51** |
| road runner | 212345.12 ± 192339.01 | **542775.42 ± 58830.61** | 0.00 ± 0.00 |
| robotank | **80.69 ± 7.32** | 50.75 ± 31.02 | 63.82 ± 20.55 |
| seaquest | 164587.95 ± 122633.80 | **310745.77 ± 67781.42** | 83490.38 ± 121434.46 |
| skiing | -29958.20 ± 4.99 | **-21898.19 ± 9898.02** | -28243.31 ± 3453.55 |
| solaris | **6037.15 ± 3103.57** | 3675.08 ± 5575.93 | 2512.84 ± 1288.33 |
| space invaders | **37497.11 ± 9973.19** | 19113.28 ± 8512.98 | 4704.35 ± 2763.78 |
| star gunner | **103870.95 ± 23124.17** | 78559.08 ± 42110.81 | 88169.92 ± 13783.74 |
| surround | 9.37 ± 0.45 | **9.40 ± 0.51** | 8.60 ± 1.00 |
| tennis | **14.08 ± 11.58** | 9.20 ± 11.29 | -0.04 ± 0.08 |
| time pilot | **36085.82 ± 6934.00** | 14250.32 ± 1651.02 | 21051.93 ± 4156.28 |
| tutankham | 33.93 ± 28.18 | 21.66 ± 13.11 | **79.51 ± 40.10** |
| up n down | 407873.64 ± 96615.12 | 237945.96 ± 101020.61 | **438679.20 ± 38137.36** |
| venture | **1754.74 ± 229.65** | 1632.54 ± 247.91 | 1624.82 ± 100.65 |
| video pinball | 248629.59 ± 231446.28 | 48305.17 ± 42432.44 | **339391.16 ± 194812.76** |
| wizard of wor | **19454.60 ± 7061.60** | 14127.73 ± 9153.21 | 18270.16 ± 12166.04 |
| yars revenge | **385904.98 ± 264624.16** | 217128.25 ± 28248.52 | 248086.47 ± 162482.95 |
| zaxxon | 20411.27 ± 12535.64 | 25063.44 ± 10385.30 | **31679.84 ± 10447.57** |

| DmLab Levels | coberl | gtrxl |
|---|---|---|
| rooms collect good objects test | $\mathbf{9.74 \pm 0.14}$ | $9.69 \pm 0.23$ |
| rooms exploit deferred effects test | $\mathbf{56.57 \pm 3.99}$ | $54.87 \pm 0.84$ |
| rooms select nonmatching object | $\mathbf{64.11 \pm 10.32}$ | $58.48 \pm 4.34$ |
| rooms watermaze | $\mathbf{56.61 \pm 1.49}$ | $51.72 \pm 3.41$ |
| rooms keys doors puzzle | $33.99 \pm 5.44$ | $\mathbf{35.53 \pm 8.33}$ |
| language select described object | $\mathbf{611.02 \pm 1.02}$ | $605.59 \pm 14.29$ |
| language select located object | $\mathbf{606.05 \pm 3.73}$ | $584.23 \pm 3.22$ |
| language execute random task | $\mathbf{232.67 \pm 28.07}$ | $198.45 \pm 15.96$ |
| language answer quantitative question | $\mathbf{327.26 \pm 4.26}$ | $297.57 \pm 9.61$ |
| lasertag one opponent large | $\mathbf{15.47 \pm 3.03}$ | $11.65 \pm 3.29$ |
| lasertag three opponents large | $\mathbf{22.99 \pm 3.79}$ | $28.46 \pm 4.59$ |
| lasertag one opponent small | $\mathbf{32.64 \pm 3.50}$ | $28.62 \pm 3.76$ |
| lasertag three opponents small | $\mathbf{42.96 \pm 3.10}$ | $34.54 \pm 0.47$ |
| natlab fixed large map | $\mathbf{46.03 \pm 8.85}$ | $43.46 \pm 14.38$ |
| natlab varying map regrowth | $\mathbf{31.07 \pm 6.74}$ | $27.62 \pm 9.09$ |
| natlab varying map randomized | $\mathbf{48.97 \pm 8.98}$ | $32.79 \pm 11.30$ |
| platforms hard | $\mathbf{58.10 \pm 7.97}$ | $35.67 \pm 23.15$ |
| platforms random | $\mathbf{85.15 \pm 1.54}$ | $69.01 \pm 1.04$ |
| psychlab continuous recognition | $\mathbf{58.91 \pm 2.17}$ | $54.96 \pm 3.87$ |
| psychlab arbitrary visuomotor mapping | $51.56 \pm 0.44$ | $\mathbf{59.96 \pm 0.33}$ |
| psychlab sequential comparison | $33.56 \pm 0.84$ | $\mathbf{35.89 \pm 0.64}$ |
| psychlab visual search | $77.96 \pm 1.33$ | $76.92 \pm 0.82$ |
| explore object locations small | $\mathbf{83.07 \pm 5.65}$ | $71.85 \pm 6.23$ |
| explore object locations large | $\mathbf{62.57 \pm 5.86}$ | $60.61 \pm 2.72$ |
| explore obstructed goals small | $2.85 \pm 5.35$ | $245.40 \pm 10.15$ |
| explore obstructed goals large | $\mathbf{109.61 \pm 1.20}$ | $79.48 \pm 6.72$ |
| explore goal locations small | $\mathbf{369.20 \pm 3.54}$ | $340.19 \pm 2.61$ |
| explore goal locations large | $\mathbf{139.74 \pm 9.84}$ | $123.20 \pm 14.65$ |
| explore object rewards few | $71.47 \pm 14.95$ | $69.02 \pm 0.95$ |
| explore object rewards many | $\mathbf{104.62 \pm 2.63}$ | $94.97 \pm 1.8$ |

# F LEARNING CURVES

## F.1 ATARI LEARNING CURVES

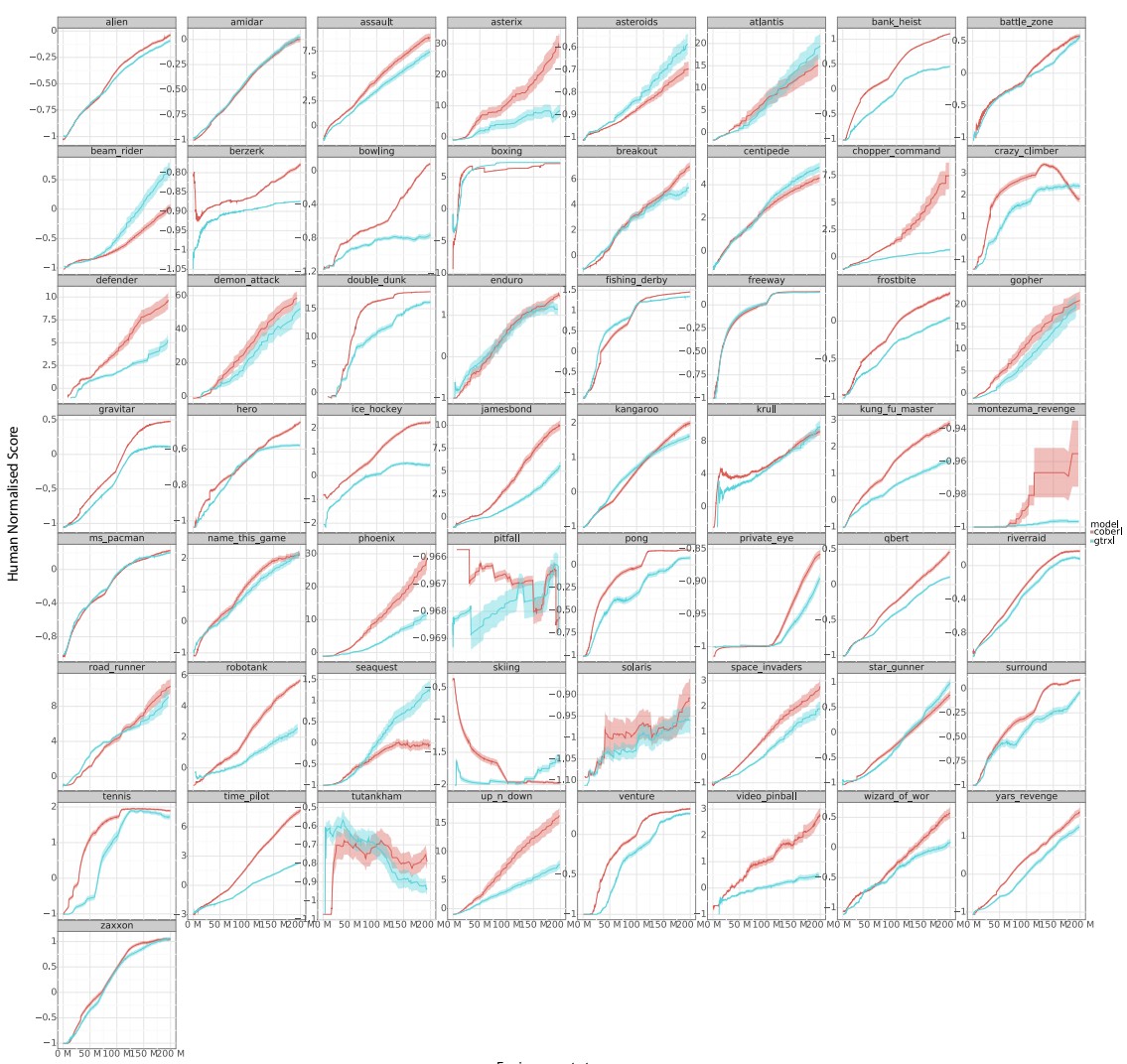

Figure 5: Learning Curves for Atari. The x-axis represents number of environment steps in millions. The y-axis represent the Human Normalised score. The error represents the 95% confidence interval. Red is COBERL, BLUE is GTrXL

## F.2 DMCONTROL LEARNING CURVES

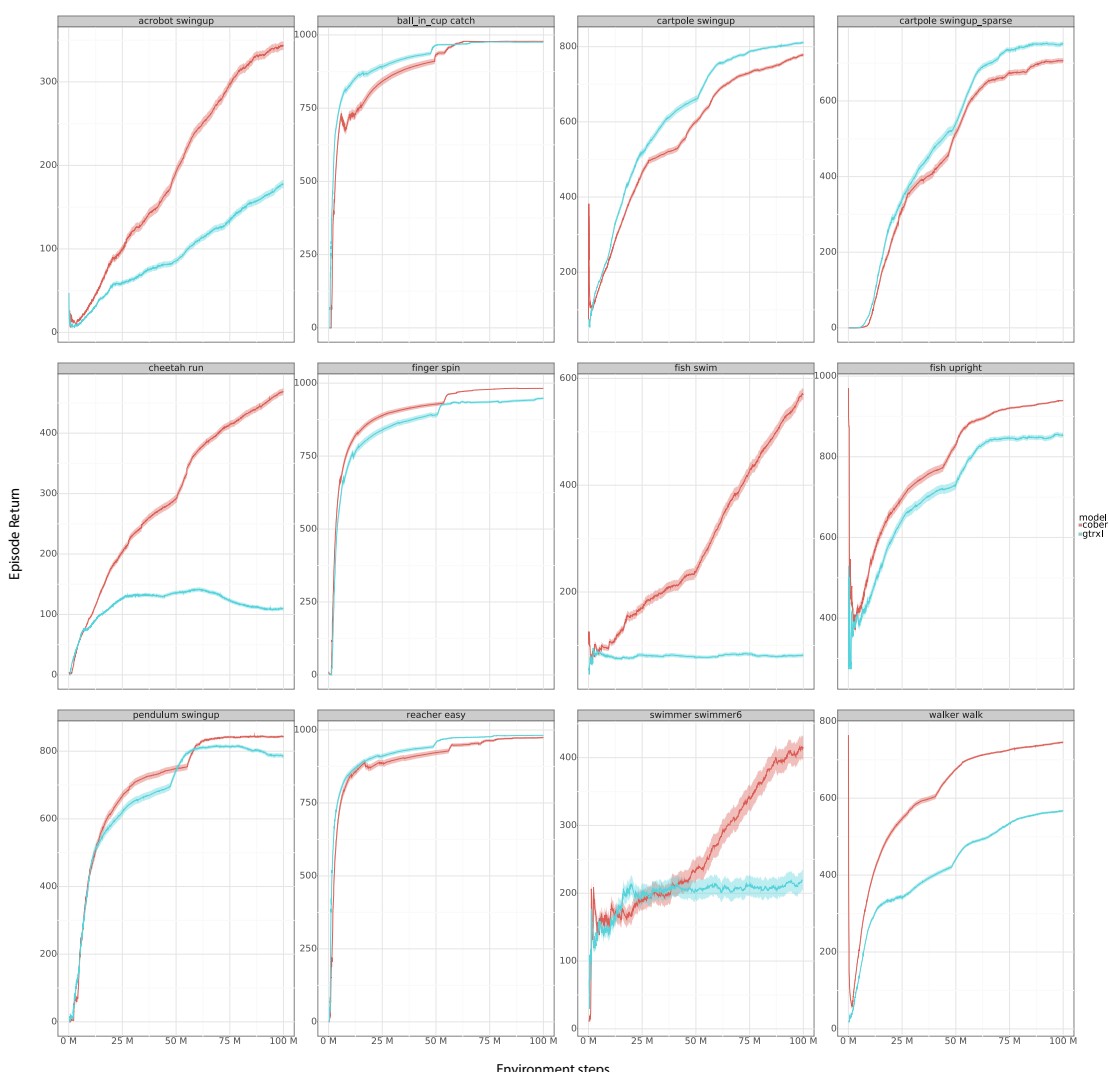

Figure 6: Learning Curves for DMControl. The x-axis represents number of environment steps in millions. The y-axis represent the Episode return. The error represents the 95% confidence interval. Red is COBERL, BLUE is GTrXL

## F.3 DMLab Learning curves

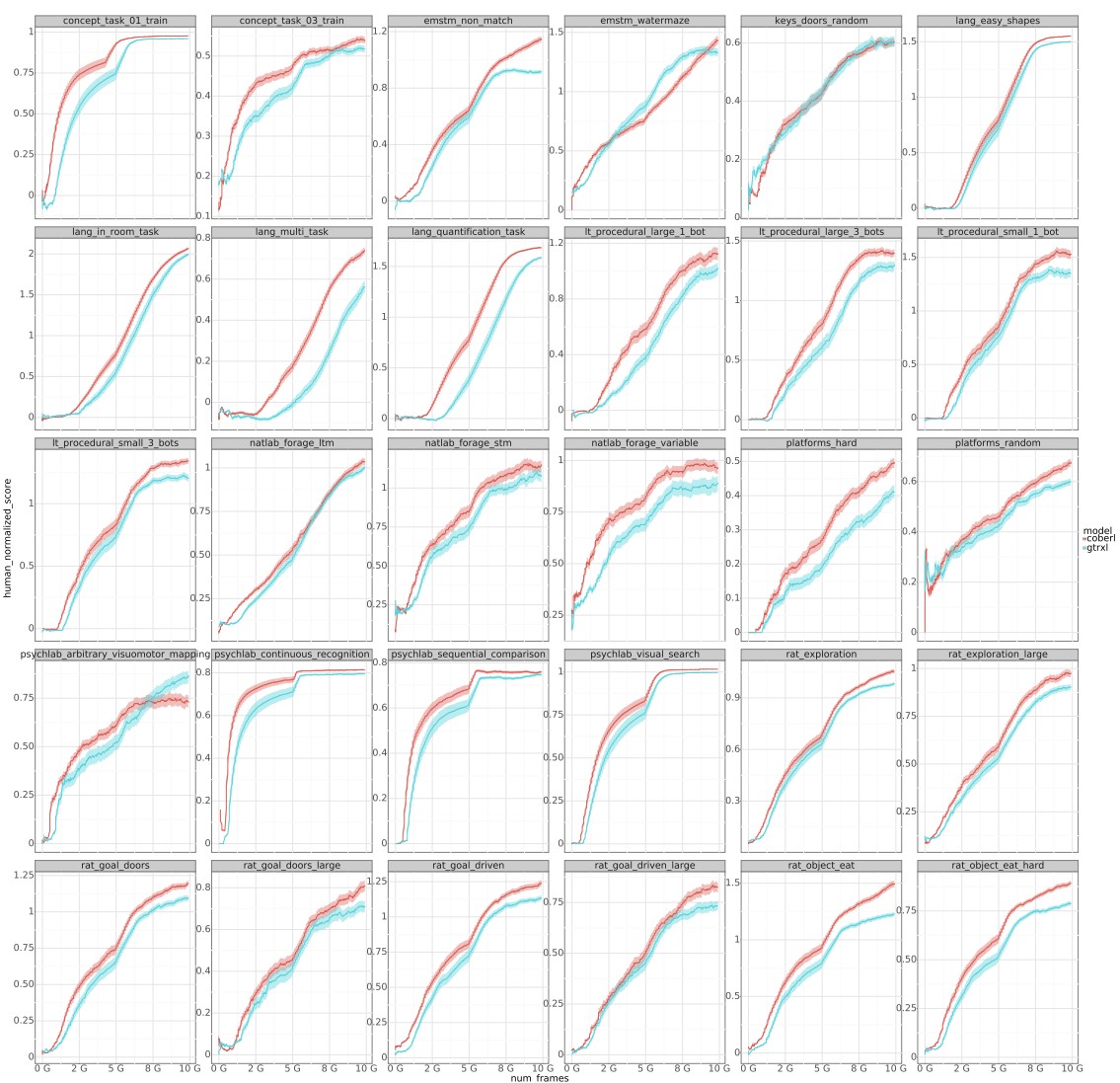

Figure 7: Learning Curves for DMLab. The x-axis represents number of environment steps in billions. The y-axis represent the Human Normalised score. The error represents the 95% confidence interval. Red is COBERL, BLUE is GTrXL

## G  LICENSES

**The The Arcade Learning Environment** Bellemare et al. (2013) is released as free, open-source software under the terms of the GNU General Public License. The latest version of the source code is publicly available at: http://arcadelearningenvironment.org

**DeepMind Control Suite** Tassa et al. (2018) is released as free, open-source software under the terms of Apache-2.0 License. The latest version of the source code is publicly available at: `https://github.com/deepmind/dm_control/blob/master/dm_control/suite/README.md`

**DmLab** Beattie et al. (2016) is released as free, open-source software under the terms of Apache-2.0 License. The latest version of the source code is publicly available at: `https://github.com/deepmind/lab/tree/master/game_scripts/levels/contributed/dmlab30`

## H  PSEUDO-CODE

---
**Algorithm 1** Pseudo-code for CoBERL
---
training_iterations ← 0
initialise_weights(VisualEncoder, TransformerStack, Gate, ValueNetwork)
**while**  training_iterations ≤ max_training_iterations **do**
    sampled_batch ← sample_experience()
    encoded_images ← VisualEncoder(sampled_batch)
    encoded_actions ← OneHotActionEncoder(sampled_batch)
    combined_inputs ← concat(encoded_images, previous_rewards, encoded_actions)
    transformer_inputs, transformer_targets, mask ← create_masks_targets(combined_inputs)
    output_transformer_contrastive ← TransformerStack(transformer_inputs, causal_mask=False)
    **contrastive_loss** ← compute_aux_loss(output_transformer_contrastive, transformer_targets, mask)
    output_transformer_RL ← TransformerStack(transformer_inputs, causal_mask=True)
    gated_output ← Gate(combined_inputs, output_transformer_RL)
    lstm_output ← LSTM(gated_output)
    combined_inputs_for_value_net ← concat(lstm_output, combined_inputs)
    value_estimation ← ValueNetwork(combined_inputs_for_value_net)
    **rl_loss** ← compute_rl_loss(value_estimation, extra_args)
    **total_loss** ← rl_loss + contrastive_loss
**end while**
---

**Pseudo-code for the auxiliary loss calculation**

```python
1  def compute_aux_loss(input1, input2, mask_ext):
2
3    batch_size, seq_dim, feat_dim = input1.shape
4    input1 = reshape(input1, [-1, feat_dim])
5    input2 = reshape(input2, [-1, feat_dim])
6
7    input1 = l2_normalize(input1, axis=-1)
8    input2 = l2_normalize(input2, axis=-1)
9
10   # Compute labels index
11   labels_idx = arange(input1.shape[0])
12   labels_idx = labels_idx.astype(jnp.int32)
13   # Compute pseudo-labels for contrastive loss.
14   labels = one_hot(labels_idx, input1.shape[0] * 2)
15   # Mask out the same image pair.
16   mask = one_hot(labels_idx, input1.shape[0])
17   # Compute logits.
18   logits_11 = matmul(input1, jnp.transpose(input1))
19   logits_22 = matmul(input2, jnp.transpose(input2))
20   logits_12 = matmul(input1, jnp.transpose(input2))
21   logits_21 = matmul(input2, jnp.transpose(input1))
22   # Calculate invariance penalty.
23   inv_penalty = kl_with_logits(   # [B * T]
24       stop_gradient(logits_11), logits_22, axis=-1)
25   inv_penalty += kl_with_logits(
26       stop_gradient(logits_12), logits_22, axis=-1)
27   inv_penalty += kl_with_logits(
28       stop_gradient(logits_21), logits_11, axis=-1)
29   inv_penalty += kl_with_logits(
30       stop_gradient(logits_12), logits_21, axis=-1)
31   inv_penalty = inv_penalty / 4.   # [B * T]
32
33   logits_11 = logits_11 - mask * 1e9
34   logits_22 = logits_22 - mask * 1e9
35
36   logits_1211 = concatenate([logits_12, logits_11], axis=-1)
37   logits_2122 = concatenate([logits_21, logits_22], axis=-1)
38
39   loss_12 = -sum(labels * log_softmax(logits_1211), axis=-1)
40   loss_21 = -sum(labels * log_softmax(logits_2122), axis=-1)
41
42   loss = reshape(loss_12 + loss_21, [batch_size, seq_dim]) * mask_ext
43   inv_penalty = reshape(inv_penalty, [batch_size, seq_dim]) * mask_ext
44
45   loss = mean(loss + inv_penalty)
46
47   return loss
```

# I   AREA UNDER THE CURVE

For all the levels we calculated the AUC by integrating composite Simpson's rule with a delta(x) of 5 steps. We use the *intergate* package from scipy (Virtanen et al., 2020).

# J   LIMITATIONS AND FUTURE WORK

A limitation of our method is that it relies on single time step information to compute its auxiliary objective. Such objective could naturally be adapted to operate on temporally-extended patches, and/or action-conditioned inputs. Also, as done in the original BERT (Devlin et al., 2019), it could be possible to add a CLS token at the beginning of each sequence sent to the Transformer and then train the CLS token with RL gradients. In this way it would be possible to directly use the embeddings

of the CLS token as a sequence summary and hence provide more context to the policy estimation network. We regard those ideas as promising future research avenues.

## K    EXTRA INFORMATION ABOUT THE GATE EMPLOYED IN EQUATION 4

The gate we use in equation 4 is derived directly from the one used in GTrXL (Parisotto et al., 2020). We report here the details for better clarity.

The gate is the defined in the following way:

$$g(y, x) = (1 - z) \odot y + z \odot \hat{h} \tag{5}$$

where:

$$z = \sigma(W_z x + U_z y - b_g) \tag{6}$$

$$\hat{h} = \tanh(W_g x + U_g(r \odot y)) \tag{7}$$

and:

$$r = \sigma(W_r x + U_r y) \tag{8}$$

$W_z, U_z, W_g, U_g, W_r and U_r$ are set of linear weights and $b_g$ is a bias.

In our case x is the output of the transformer network and y is the output of the encoder network.

## L    EXTRA ANALYSIS

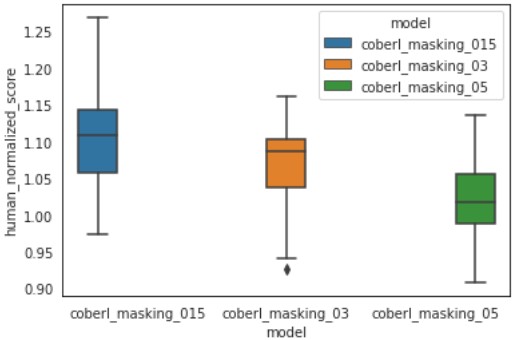

Figure 8: Human normalised score as a function of the masking percentage on the input sequence. $coberl\_masking\_015$ relates to $15\%$ masking of the input sequence, $coberl\_masking\_03$ relates to $30\%$ masking of the input sequence and $coberl\_masking\_05$ relates to $50\%$ masking of the input sequence.

We also attempted higher masking rate, but as shown in figure L they seem to perform worse, probably because high level of masking are reducing to much the number of frame present in the sequence, hence removing the information need to successfully perform the auxiliary task.

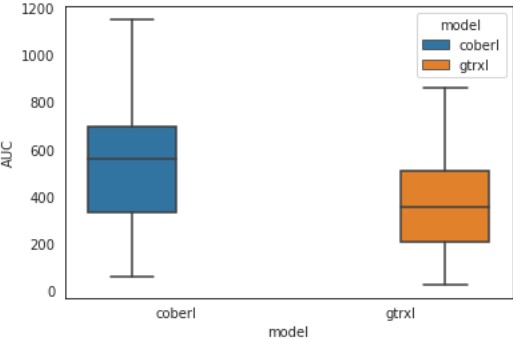

Figure 9: AUC calculated at 100% human score.

Even when calulcate at 100% human score, the AUC shows a significant advantage of CoBERL over gTrXL t(60)=2.616, p=0.0011. CoBERL M=545.18, SD=282.58; gTrXL M=374.49, SD=282.58.

