# OpenReview forum: "CoBERL: Contrastive BERT for Reinforcement Learning"
_ICLR.cc/2022/Conference — ICLR 2022 Spotlight_

### Official Review · Reviewer_YoL5 · 2021-11-02

**Correctness:** 3
**Technical Novelty And Significance:** 2
**Empirical Novelty And Significance:** 3
**Recommendation:** 6
**Confidence:** 3

**Main Review:**

Normally when I hear "LSTM" I think "hard to train, requires many samples". When I hear "Transformer" I think the same thing. Yet in this work it is being claimed that a combination of these data-hungry methods results in higher data efficiency that with other methods. At which point I naturally get suspicious :) and want to investigate more.

Major concerns / criticisms / discussion
----------------------------------------

* The abstract promises that "COBERL consistently improves data efficiency across the full Atari suite, a set of control tasks and a challenging 3D environment", however most of the reported results are about performance improvement (after 200M steps (on Atari)). The results on improved data efficiency come in a for of AUC curves as compared to GTrXL. My understanding is that GTrXL was chosen as baseline because it demonstrated the highest human normalized score, but if the main question is about data efficiency and AUC, then a method that rises quickly (Rainbow?) and plateaus later would have a higher AUC, especially at an earlier cut-off points (for example 50M instead of 200M). What if we change the metric from "AUC at 200M steps" to "AUC at the moment when an algorithms achieves 100% human performance"? The motivation behind this line of thinking comes from domains where data samples are expensive (real robots, medical observations, etc) where what we care about is getting to good performance fast, and not so much about "being more data efficient by the time we have collected 200M data samples". Do we know that GTrXL is a good baseline for data efficiency even if we discard the arbitrarily chosen (by the community) mark of 200M samples?

* The concept of "self-attention consistency" that seems to be a crucial idea driving this contribution seems underexplained to me: what do authors mean by self-attention being consistent, consistent across what, and why that consistency is a good thing?

* A subjective note on readability: This paper was quite a hard read, perhaps because the explanation of the introduced concept and main ideas is mixed with implementational details and minor notes throughout the text. This makes it a bit hard to distill the core idea and important parts of the contribution from non-essential implementational details.


Questions
---------

* In the Rainbow paper (https://arxiv.org/pdf/1710.02298.pdf) the reported improvement over human is ~200%+ (Figure 1), in the current work it is reported as 874%, where does this difference come from?

Minor remarks
-------------

page 3 "augmentations make other methods less data efficient" -- unclear what is meant by this statement, isn't the point of augmentations the opposite?

page 7 par 1: typo "Figure 2 presentt this analysis"

Figure 2 caption "els.(higher is better)" -- some formatting issue with spaces and the dot

**Summary Of The Paper:**

The paper replaces the recurrent architecture of R2D2 with a Transformer-LSTM hybrid, and introduces a bi-directional contrastive loss that will be combined with the RL loss for overall optimization. The architecture is supplemented with a learnable gate between the Transformer and LSTM layers that should learn to regulate whether Transformer's output at time t should be fed into LSTM or it is not necessary, making the overall learning easier and leading to higher data efficiency and performance than the baseline (GTrXL).

**Summary Of The Review:**

My evaluation largely depends on whether the chosen beaseline is consistent with the stated main contribution of this work: would AUC of other methods for RL would also be lower at earlier cut-off points. I see two potential possibilities:
(a) The chosen baseline is actually good for data efficiency -> the main claim of the paper holds, all's good
(b) The baseline was not good for data efficiency -> the paper should not claim data efficiency, but rather just go with performance increase (1425%, while baseline is 1202%).
I am looking forward to the discussion to gain clarity on this point.

---

> ### Author Response · Authors · 2021-11-17
> **Clarifications**
>
> We thanks the reviewer for the in depth review of the paper and the time spent on it.
> Below we address the points raised.
>
> > The abstract promises that "COBERL consistently improves data efficiency across the full Atari suite, a set of control tasks and a challenging 3D environment", however most of the reported results are about performance improvement (after 200M steps (on Atari)). [...]
>
> We thank the reviewer for raising this point. As they mention, we do use the 200M frame         benchmark that is most commonly used in literature and accepted by the community. While methods like rainbow would have higher data efficiency if we limit ourselves to reaching the human benchmark and measure efficiency as the number of frames or AUC to reach the human benchmark score, we believe that 1) as highlighted in the last sentence of the abstract, one of the main outcomes of the work is indeed the combination of data efficiency with strong final performance 2) there is great value in improving data efficiency for large distributed models that achieve state-of-the-art performance. We have refined our wording in the abstract to reflect this.
>
> Also, we believe that the proposed metric (auc at 100%), whilst interesting, does not add much beyond our existing metrics. For example, we performed this analysis  on DmLAB (appendix L) because in this domain collecting samples is expensive, as rendering several billion of 3D frames is expensive also on a GPU. However, as it can be seen in appendix L, the results concur with our existing metrics.
>
> > The concept of "self-attention consistency" that seems to be a crucial idea driving this contribution seems underexplained to me: what do authors mean by self-attention being consistent, consistent across what, and why that consistency is a good thing?
>
> The idea of self-attention consistency is based on the fact that, with self-attention, a sequence is analyzed in parallel, that is all the time steps are considered at the same time. This parallel computation, by considering all the frames at the same time, it helps the agent in taking into account a basic fact about the world: the composing parts of a scene cannot simply go in and out of existence over time. For instance, if an object is present on the scene and then it disappears at one location, and it immediately appears at a different (spatially separated) location, then those two instances cannot be the same object. This is particularly important for our auxiliary loss as it has been shown that the combination of self-attention and masking  forces the model to focus on the context provided by the surrounding timesteps to solve its objective [1] . Thus, when the model is asked to reconstruct a particular masked frame it does so by attending to all the adjacent frames in the trajectory, and this will help maintaining consistency  over time. This explanation has now been included in section 2.1
>
> [1] Voita, Elena, Rico Sennrich, and Ivan Titov. "The bottom-up evolution of representations in the transformer: A study with machine translation and language modeling objectives." arXiv preprint arXiv:1909.01380 (2019).
>
> > A subjective note on readability: This paper was quite a hard read,
>
> Thanks to the reviewer for this feedback. In accordance with the feedback that we got from other reviewers we re-wrote a large part of section 2 and so we hope that in this way the paper is more readable.
>
> > In the Rainbow paper (https://arxiv.org/pdf/1710.02298.pdf) the reported improvement over human is ~200%+ (Figure 1), in the current work it is reported as 874%, where does this difference come from?
>
> In the Rainbow paper the improvement over the human benchmark is ~200% on the median human normalized score, 874% is the improvement over the mean human normalised score, not the median. These scores are reported in our Table 1.
>
> > page 3 "augmentations make other methods less data efficient" -- unclear what is meant by this statement, isn't the point of augmentations the opposite?
>
> Thank you for catching this statement which would be commonly understood to be intuitively the opposite. The intent behind the statement is to highlight that more data (the augmentations) needs to be processed by the agent. We have refined the wording to reflect that.
>
> > page 7 par 1: typo "Figure 2 present this analysis"
>
> Fixed
>
> > Figure 2 caption "els.(higher is better)" -- some formatting issue with spaces and the dot
>
> Fixed

---

> > ### Comment · Reviewer_YoL5 · 2021-11-29
> > **Post-rebuttal**
> >
> > Thank you for clarifying some of the points I've raised in my review.
> > The validation of the claims about data efficiency still do not appear convincing to me:
> > * firstly due to the arbitrary cut-off point where this efficiency is measured, and
> > * secondly due to the fact that AUC measure is pulled over all games and averaged together, leaving it a bit unclear about the specific effects of the method on data efficiency
> >
> > The overall performance on the other hands is shows to overcome the chosen baseline and does provide us with another improvement on Atari games.
> >
> > Taking into account the curious architecture I find the paper is worth sharing with the community, and keeping my original score.

---

### Official Review · Reviewer_XmhC · 2021-11-02

**Correctness:** 3
**Technical Novelty And Significance:** 3
**Empirical Novelty And Significance:** 3
**Recommendation:** 8
**Confidence:** 4

**Main Review:**

Strengths

* The problem of the growing size of transformers as we require larger time dependencies is of great concern, the trick of forwarding these representations to an LSTM layer is simple yet effective from the results provided
* Avoiding the need of hand-engineered data augmentations is a big plus for representation learning subobjectives
* The paper evaluates the proposed solution in multiple hard and well known benchmarks, authors also provide relevant ablation studies

---
Weaknesses

* General comments:
   * My biggest concerns are about clarity and presentation, the paper relies heavily on readers previous knowledge, which requires combined combined intuition from multiple disciplines. Additionally, several points of the architecture are never explained (neither given a intuition to the reader). Below I provide specific details
   *  The number of random seeds in the empirical evaluation, only 3, is below current accepted standard for accepted RL works (which is a minimum of 5).
---
Below I detail specific points that I believe need further clarification (in order of appearance):

* I missed a brief background section explaining how representation learning is used in RL, also, this paper gets and improves ideas from ReLIC, we should be able to understand this work without requiring go to ReLIC's
* Section 2.1, 3rd paragraph, you mention that 15% of the embedding is masked, it seems a very specific value, which ones did you explore? what happened with those different values? (some intuition)
* Also there, and in section 2.1 in general, I missed to be given intuitions of why each component was designed in that specific way. E.g., starts directly explaining the masking, I would try to build into the reader the intuition of  why are we masking? before entering in the details. This is much better done in section 2.2 for instance.
* What is exactly doing the "Gate" function? it is mentioned multiple times, and it is evaluated in the ablations, but I couldn't find where is it explained what is this function. It is mentioned that helps to skip the transformed output until the weights are warmed-up, how does it knows this?. I believe that the answer to this would also fit well in a background section.
* Related work section, paragraph 2, says "Second, CoBERL combines the transformer architecture with GRUs to..." until this point GRUs are never mentioned, only LSTMs, so it is not evident if you use both indistinctively (if that is the case I would only say recurrent layer) or if not, where are the GRUs in the architecture and what is their role.
* First paragraph page 6, second to last line, you say "We use Peng's Q($\lambda$) " this is missing a proper citation.
* Small typo, two lines below what I mentioned above, you are missing an space after V-MPO

----------
### Post Author's response and updated version

I think that authors did a good work in the updated version, the work is much better self-contained now and easier to follow. Importantly, now all the experiments are the result of 5 independent runs instead of 3, which raises the reader's confidence in the empirical results.

Given the strengths I reported in my original review, I updated my scores.





**Summary Of The Paper:**

The paper introduces a new transformer architecture, called CoBERL, targeting reinforcement learning. The key ingredients of the proposed architectures are the use of LSTMs to reduce the need of large transformers and a contrastive learning procedure that doesn't require of human data augmentation.

**Summary Of The Review:**

The new architecture is interesting, proposes novel ideas that solve important issues of previous solutions and shows good empirical results. However, the paper has clarity issues and doesn't seem completely self-contained. Also the reduced number of random initializations goes below usual quality standards in published RL work.

Still, I believe these points could be fixed through the rebuttal, so I am open to revising my score if the authors submit an updated version that tackles these points.

----------
Authors correctly addressed my concerns in the updated version

---

> ### Author Response · Authors · 2021-11-17
> **Extra seeds and clarification**
>
> We would like to thank the reviewer for their thoughtful feedback. We really appreciate that the reviewer critically highlights the contribution of our work. Below we detail our answer to the review concerns.
>
> > My biggest concerns are about clarity and presentation, the paper relies heavily on readers previous knowledge, which requires combined combined intuition from multiple disciplines. [...]
>
> We agree with the reviewer and so we re-wrote section 2.1 in accordance with the comments provided below. We also made sure to make the paper more self-contained by briefly describing ReLIC in the re-written section 2.1. Also, we want to emphasize that the loss in CoBERL is derived directly from ReLIC, so considering the limited space, equation 3 and the pseudo-code in appendix H could be considered good summaries of ReLIC paper. Although using the same loss as ReLIC, CoBERL differ from it in 2 key ways:
> * does not use data augmentations, but it relies on the sequential nature of the input data to create the positive and negative samples used in the loss.
> * we do not use an additional encoder network , thus making CoBERL fully end-to-end. An aspect that simplifies the approach and reduces the parameters compared to the original ReLIC.
>
> Similarities and differences between the 2 approaches have now been clarified in the paper.
>
> > The number of random seeds in the empirical evaluation, only 3 [...]
>
> We run 2 extra seeds for all our experiments in Atari, DmControl, and DmLab and all the ablations. We updated the tables and the plots in accordance with these new results. The conclusion of the paper did not change, but we agree with the reviewer that in this way the paper is more solid.
>
> Below we report specific answers to the specific points raised by the reviewer.
>
> > I missed a brief background section explaining how representation learning is used in RL [...]
>
> We expanded section 2.1 to include a background paragraph which discusses the why and how representation learning is used in RL. Also we now provide a brief explanation of ReLIC and a clear remainder to the ReLIC pseudo-code in the appendix.
>
> > Section 2.1, 3rd paragraph, you mention that 15% of the embedding is masked, it seems a very specific value, which ones did you explore? what happened with those different values? (some intuition)
>
> We now better explain why masking is important and we also added a section in the appendix L that gives some intuitions on extra values. 15% was the number taken from previous work in language modelling [1]. We also attempted higher masking rate, but as shown in appendix L they seem to perform worse, probably because high level of masking are reducing to much the number of frame present in the sequence, hence removing the information need to successfully perform the auxiliary task.
>
> [1] Devlin, Jacob, et al. "Bert: Pre-training of deep bidirectional transformers for language understanding." arXiv preprint arXiv:1810.04805 (2018).
>
> > Also there, and in section 2.1 in general, I missed to be given intuitions  [...]
>
> We now provide intuitions for both masking and contrastive learning, before describing the details of the loss. Thus making section 2.1 closer to 2.2
>
> > What is exactly doing the "Gate" function? [...]
>
> The gate function was taken directly from the GTrXL paper, we now make this more explicit and we also report the related math in the appendix K such that the read can have a more self-contained paper. Regarding the skip of the transformer output the idea is that the gate can decide how much weighting to give to each of its two inputs, that is i) the transformer output and ii) the encoder output. Given that the gate is trained directly using the RL loss then the weighting should be proportional to what is best to achieve a better prediction of the value function. Also, it has been shown in recent work that a similar gate technique is useful to warm up transformer networks [1], [2], we included these remainders.
>
> [1] ​​He, Kaiming, et al. "Delving deep into rectifiers: Surpassing human-level performance on imagenet classification." Proceedings of the IEEE international conference on computer vision. 2015.
>
> [2] Jumper, John, et al. "Highly accurate protein structure prediction with AlphaFold." Nature 596.7873 (2021): 583-589. (Supplementary material)
>
> > Related work section, paragraph 2, says "Second, CoBERL combines the transformer architecture with GRUs to..." until this point GRUs are never mentioned [...]
>
> This is related to the above point. Given that the gate is a derivation of the gate mechanism used in the GRU we used this term. However, we understand this is causing confusion and so we clarified it.
>
> > First paragraph page 6, second to last line, you say "We use Peng's Q(λ) " this is missing a proper citation.
>
> We added it, thanks for spotting this
>
> > Small typo, two lines below what I mentioned above, you are missing an space after V-MPO
>
> Thanks, fixed.

---

> > ### Comment · Reviewer_XmhC · 2021-11-19
> > **Thank you for your answers I believe that now it is in much better shape**
> >
> > I want to thank the authors for responding all my doubts and concerns.
> >
> > I have been carefully checking the additions in the  updated version and I believe that the paper is easier to follow now (Section 2.1 is definitely much easier to follow now). The additional seeds also raise the strength of the empirical analysis.
> >
> > I am updating my scores and recommendation accordingly

---

> > > ### Author Response · Authors · 2021-11-19
> > > **Thanks.**
> > >
> > > We would like to thank the reviewer for the time spent on our paper again.
> > > We are glad to hear that the modifications we made to the paper make it easier to read, the feedback we received from the reviewer was absolutely instrumental to achieve this.
> > >
> > > Thanks for updating the score.
> > >
> > > Best regards,

---

### Official Review · Reviewer_gQY1 · 2021-11-02

**Correctness:** 4
**Technical Novelty And Significance:** 3
**Empirical Novelty And Significance:** 3
**Recommendation:** 8
**Confidence:** 3

**Main Review:**

The proposed approach seems like a novel combination of existing methods with the advantage of requiring less prior knowledge and higher sample efficiency.
Using time steps from other trajectories in the buffer seems well motivated and avoids the need for designing augmentations.
Experiments are comprehensive and the ablation study is appreciated.
A minor gripe is the use of Y for input and X for output of the transformer.

**Summary Of The Paper:**

The paper introduces a contrastive learning objective to learn representations for an RL agent. The basic objective is taken from ReLIC, but instead of augmenting states (which requires prior knowledge of the problem domain) for each timestep the authors use timesteps of the other trajectories in a minibatch as negative examples.

**Summary Of The Review:**

The proposed approach is a novel combination of existing methods that is well motivated. Experiments are comprehensive, including an ablation study.

---

> ### Author Response · Authors · 2021-11-17
> **Thanks for the review**
>
> We would like to thank the reviewer for his positive feedback. In particular we appreciate that they touched on all the relevant contributions of our work, in particular the ability of our methods to avoid the use of data augmentations. Thanks also for highlighting the experimental section with the inclusion of our set of ablations.

---

> > ### Comment · Reviewer_gQY1 · 2021-11-30
> > **Thanks for your answers**
> >
> > While other reviewers raised some concerns about clarity I still feel that these are not severe enough to prevent the publication of this paper. Especially after the changes made during the rebuttal I stand by my score.

---

### Public Comment · ~Rishabh_Agarwal2 · 2021-11-09
**Statistical uncertainty in aggregate metrics**

Dear authors,

I noticed that statistical uncertainty estimates are only reported for mean and not for median scores / percentiles etc on Atari results. However, their is a sizeable uncertainty in median scores [1] on Atari 2600 games for prior algorithms such as DreamerV2 and Rainbow and it should be reported for reliable evaluation. A simple way to do so is to use stratified bootstrap confidence intervals [1]. You can easily do so using the library at https://github.com/google-research/rliable or the [colab notebook](https://bit.ly/statistical_precipice_colab).

Regarding aggregate metrics, mean could be easily dominated by outliers while median remains unchanged even when  scores on half of the task is set to 0. A better metric might be interquartile mean across all runs and tasks. Additionally, performance profiles would capture variability in performance across tasks and runs and percentiles/ aggregate metrics can be easily read from such profiles.

[1] Agarwal, R., Schwarzer, M., Castro, P.S., Courville, A. and Bellemare, M.G., 2021. Deep reinforcement learning at the edge of the statistical precipice. In NeurIPS.

---

### Decision · Program_Chairs · 2022-01-20

**Decision:**

Accept (Spotlight)

**Comment:**

This paper introduces a new transformer architecture for representation learning in RL. The key ingredients of the proposed architectures are a novel combination of existing methods: (1) the use of LSTMs to reduce the need for large transformers and (2) a contrastive learning procedure that doesn't require human data augmentation.  The resulting approach requires less prior knowledge and provides higher sample efficiency. The paper is convincing, with comprehensive experiments on multiple challenging and well-known benchmarks and an ablation study. The reviewers did expressed concerns that parts of the paper are a very difficult read and could use improvement, especially those relying on substantial external background. The intuition behind several components could be improved, and there are some clarity issues, as detailed in the individual reviews.